# Data-In-situ Computing with One-Pixel-Multiple-Memristor Architecture for Neuromorphic Sequential Vision

Yi Sun[1,2,5], Peiwen Tong[1,2,5], Jiangrong Shen[3,5], Hui Xu[1,2], Rongrong Cao [1,2], Chang Liu[1,2], Changlin Chen[1,2], Bing Song[1,2], Yinan Wang[1,2], Wei Wang [1,2] ✉, Yuchao Yang [4] ✉ & Qingjiang Li [1,2] ✉

Neuromorphic vision systems based on memristors offer an energy-efficient approach to artificial vision, yet traditional pixel(s)-to-one-memristor architectures remain inefficient in dynamic image processing due to limited temporary storage. Here, inspired by human visual working memory, we propose a one-pixel-multiple-memristor (1PnR) architecture with a rolling exposure strategy for fast sequential image acquisition. Furthermore, a data-in-situ computing network for efficient image processing is developed. With network weights mapped to voltage vectors and applied to the image storage memristor array, direct computation is enabled where the image is stored, and the energy-intensive data transmission is eliminated. A hardware prototype of the 1PnR architecture achieved 95.7% recognition accuracy on the Weizmann human action flow dataset. Compared to CMOS-based systems, this architecture is estimated to have a 2000× reduction in latency for image sensing and storage, and a 160× reduction in energy consumption image processing, demonstrating significant potential for future neuromorphic visual systems.

The human visual system is distinguished by its remarkably energy efficiency[1]. Studying and emulating the human visual system offers a promising approach to developing high-performance and low-power artificial vision systems[2,3]. Figure 1 illustrates the working process of the human visual system. In essence, the human visual system can be broadly divided into two key components: the retina and the brain. The retina cells convert light signals into neural spiking signals, which are then transmitted to the brain via the optic nerve. In the brain, synaptic cells process these spiking signals, enabling humans to visually perceive the external environment[4]. It is reported that there is a visual working memory mechanism in the brain, which stores temporary visual information and performs a pre-processing for the following neural classification[5-7]. This working mode can greatly reduce the

visual transmission in the brain nerve and further improve the process efficiency.

Recently, various artificial vision systems imitating the biological visual processing mechanism have been proposed[8-16]. Among them, memristors are promising candidates for visual information storage and in-memory neuromorphic processing[17-19]. The memristor based neuromorphic architecture can avoid the constant data conversion and transmission between physical separated image sensors and processing units in von Neumann architectures, thus provides high power and time efficiency[20-23]. In the current implemented systems, images are captured by sensors or optoelectronic devices and transmitted to memristor arrays that store weights for network computation[24-27]. While this analog-domain architecture effectively improves processing

[1]College of Electronic Science and Technology, National University of Defense Technology, Changsha 410073, China. [2]Hunan Key Laboratory of Aerospace Intelligent ASIC Technology, Changsha 410073, China. [3]Faculty of Electronic and Information Engineering, Xi'an Jiaotong University, Xi'an 710049, China. [4]Guangdong Provincial Key Laboratory of In-Memory Computing Chips, School of Electronic and Computer Engineering, Shenzhen Graduate School, Peking University, Shenzhen 518055, China. [5]These authors contributed equally: Yi Sun, Peiwen Tong, Jiangrong Shen. ✉e-mail: wangwei_esss@nudt.edu.cn; yuchaoyang@pku.edu.cn; qingjiangli@nudt.edu.cn

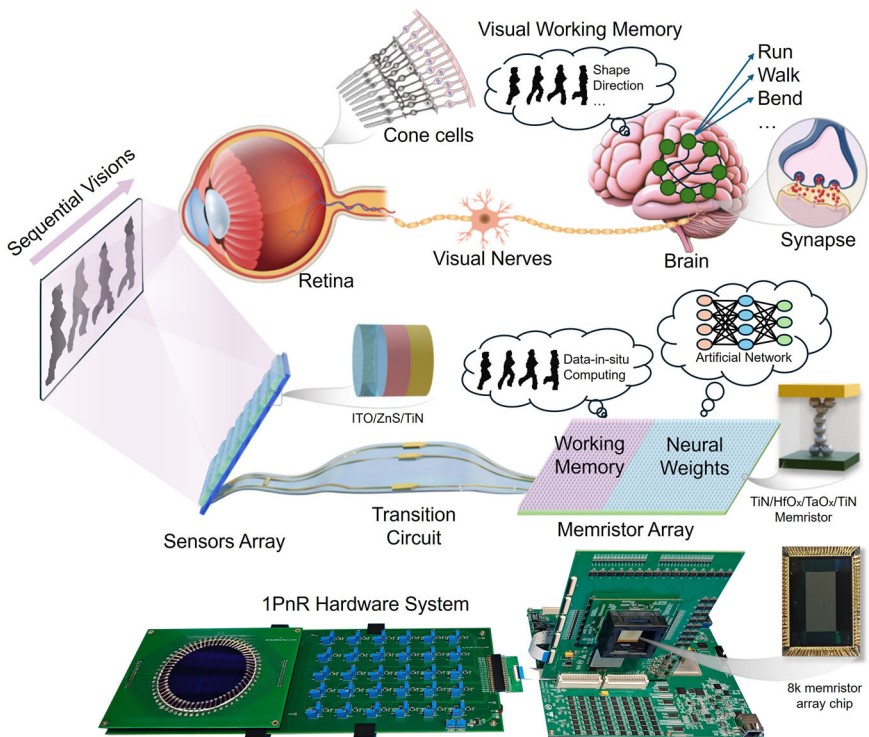

**Fig. 1 | Schematic of the human visual system and the proposed 1PnR visual system.** The human vision system contains eyes, optic neurons and the brain. The visual working memory in the human brain stores temporary sequential visual information and performs pre-processing. The proposed artificial visual system based on 1PnR architecture is composed of ITO/ZnS/TiN pixel sensors for image sensing, an analog circuit for data transmission, and an 8k memristor array based on TiN/HfO$_x$/TaO$_x$/TiN stack. The array is divided into two parts; one part serves as working memory, which stores image data and performs data-in-situ computing for pre-processing. The other part serves as a neuromorphic computing core, which stores network weights and carries out network computing for classification.

efficiency, it still requires the transmission of image data from the sensing-storage array to the computing array, resulting in additional latency and power consumption. In addition, the sensor and memristor device are usually integrated as one-sensor-one-memristor[28–30], or multi-sensor-one-memristor structure[31,32]. In sequential image processing conditions, read-out and erasing operation is required for previously stored image data before sensing new pixel data, which limits the image sensing speed and energy efficiency.

In this work, inspired by the high-efficiency working mode of the visual nerves in the human system, we propose a neuromorphic visual architecture, namely, one-pixel- multiple-memristor (1PnR) computing architecture for sequential image sensing, storing and processing, as shown in Fig. 1. The light sensors and memristor array, similar to biological retina, are connected by analog circuits as optic nerves for conversion and transmission of the sensory data. The array is divided into 2 parts, one part serves as visual working memory, which stores image data and performs data-in-situ computing for pre-processing. The other part serves as a neuromorphic computing core mimicking the visual cortex, which stores network weights and carries network computing for classification. Furthermore, the sensors and memristor array are integrated by a gate multiplexing architecture, where a sensor is connected to a gate line of the 1T1R array, offering the ability to store a single channel of optic input to multiple memristor cells simultaneously, functioning as retinal divergent connectivity for visual information processing. Based on the structure, a rolling exposure strategy for fast sequential image acquisition is then proposed. With the pixel image sensed and stored column by column to the memristor array in the analog domain, both time and energy efficiency can be achieved compared to a traditional CMOS system. Moreover, a data-in-situ computing network is proposed for fast image processing. Voltage vectors carrying network weights are applied to the image storage array to implement data-in-situ computing, which avoids transmission of the image data in traditional memristive neuromorphic systems. Finally, a 1PnR hardware system is established for verification of sequential visual sensing and processing. The experimental results reveal 95.7% recognition accuracy for the Weizmann human action flow dataset, demonstrating the great classification capacity of the data-in-situ network. Compared to typical CMOS-based systems, the proposed architecture is estimated to have ~2000 times reduction of time latency for image sensing and storage, and ~160 times reduction of energy consumption for image processing. These results demonstrate great potential for the proposed 1PnR novel architecture to construct neuromorphic visual systems.

## Results

### Device characterization and the one-pulse modulation method

A 1T1R memristor array with 8k scale (64 rows × 128 columns) is deployed in this work. Each cell contains a TiN/TaO$_x$/HfO$_x$/TiN memristor and a transistor connected in series, as shown in Fig. 2a. The schematic diagram of the 1T1R structure, as well as the transistor's output characteristic are presented in Supplementary Fig. 1. Figure 2b is the typical quasi-DC curve of the 1T1R structure. With a 1.6 V gate voltage in the SET voltage sweep (0 to 1.0 V) and a 3.5 V gate voltage in the RESET sweep (0 to −1.15 V), a resistive switching (RS) window, defined as the ratio between the high resistance state (HRS) and the low resistance state (LRS), can reach ~10. Electric pulses are then applied to the 1T1R cell, pulse trains with gradually rising amplitudes (0.8–1.3 V for SET and 1.3–2.4 V for RESET, width time 80 ns) are used as the modulation signal to achieve more resistance states and a larger switching window, and the gate voltage is set to 3 V to turn on the transistor. The result in Fig. 2c shows the continuous change of the device state under SET and RESET operations, with 100 pulses each.

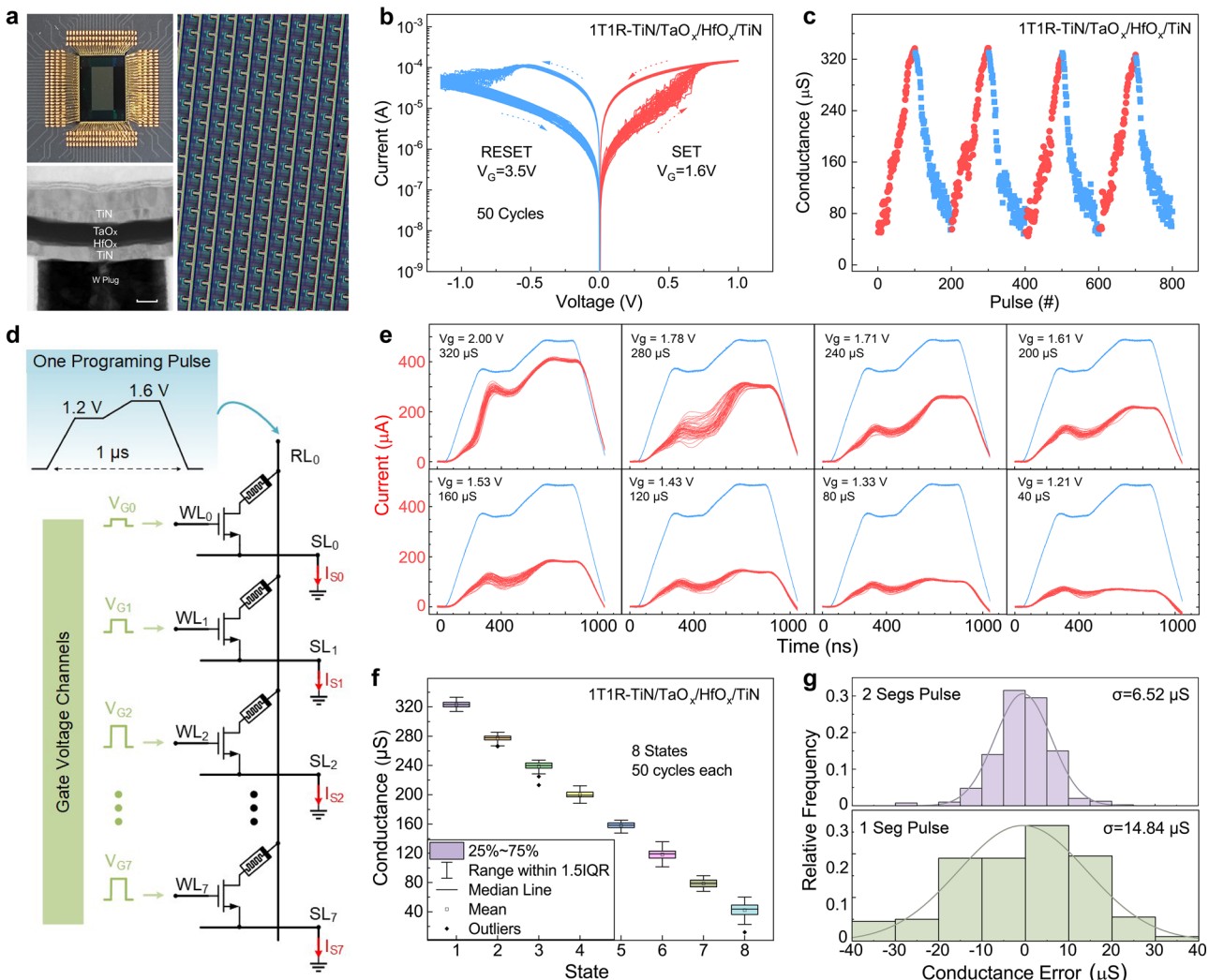

**Fig. 2 | The 1T1R memristor array and one-pulse-modulation method. a** Optical image of the 1T1R memristor chip and the TEM image of the TiN/TaO$_x$/HfO$_x$/TiN memristor structure. (Scale bar: 50 nm). **b** The quasi-DC characteristic of TiN/TaO$_x$/HfO$_x$/TiN memristor. The device shows reversible resistive behaviors with good uniformity. **c** Hundred conductance states of the 1T1R cell modulated by pulse train, indicating the great multilevel feature. **d** The design of one-pulse-modulation. The pulse is designed with two levels to increase device stability; eight devices in the same RL can be operated by one pulse simultaneously. **e** The waveform of the one-pulse modulation test for 8 devices with various gate voltages for different states, each state is performed for 50 cycles. **f** The conductance distribution of OPM. Eight distinguishable levels can be achieved, indicating the potential for fast data storage. **g** Comparison of the two-segment pulse and one-segment pulse modulation, where the two-segment pulse shows a narrower distribution of the conductance error with better modulation stability.

After each pulse, a small pulse of 0.1 V follows behind to read the device conductance. Indeed, the pulse parameters (amplitude and width) are optimized through multiple experiments to achieve better resistive-switching window and linearity. The results reveal that the 1T1R cell has analog switching behavior under pulse modulation, demonstrating its great potential in multi-value image storage and neuromorphic computing applications.

The pixel sensor is achieved by an ITO/ZnS/TiN photo-resistive structure. The top electrode is deposited with transparent ITO. ZnS is deposited as a functional film due to its excellent photoactivity. A UV LED with a wavelength of 365 nm is used to investigate optoelectronic characteristics. The experimental results plotted in Supplementary Fig. 2 show that the device has great cycle-to-cycle (C2C) and device-to-device (D2D) uniformity, with the ability to sense the light pulse with various intensities, which can effectively support the implementation of the proposed 1PnR system.

To achieve high-frame processing of sequential visual information, an open-loop fast modulation method is then proposed for the

1T1R cell, aiming to significantly improve the writing speed while partially sacrificing precision. This method applies only one parameter-fixed pulse to the 1T1R cell, which can be called one-pulse modulation (OPM). As shown in Fig. 2d, the pulse is applied to the RL, and the SLs are all grounded. Based on the inter-connection of the 1T1R array, all the devices in the same RL can be modulated at the same time, and different gate voltages can be applied to control the modulation states in different SLs. The total pulse width is 1 μs, corresponding to a theoretical modulation frequency of 1 MHz. In our experiments, eight gate voltage values are used as representatives to evaluate OPM capability and writing fluctuation. The corresponding current response of the 1T1R cell is shown in Fig. 2e, 8 devices with different gate voltages are repeatedly tested 50 times. Before each modulation, the devices are initialized to the same state (30 μS). As the gate voltage gradually drops from 2.0 V to 1.21 V, the response current of the device also gradually decreases. The current fluctuation is relatively large at the initial stage of the pulse and becomes concentrated at the pulse falling edge, indicating that the writing state

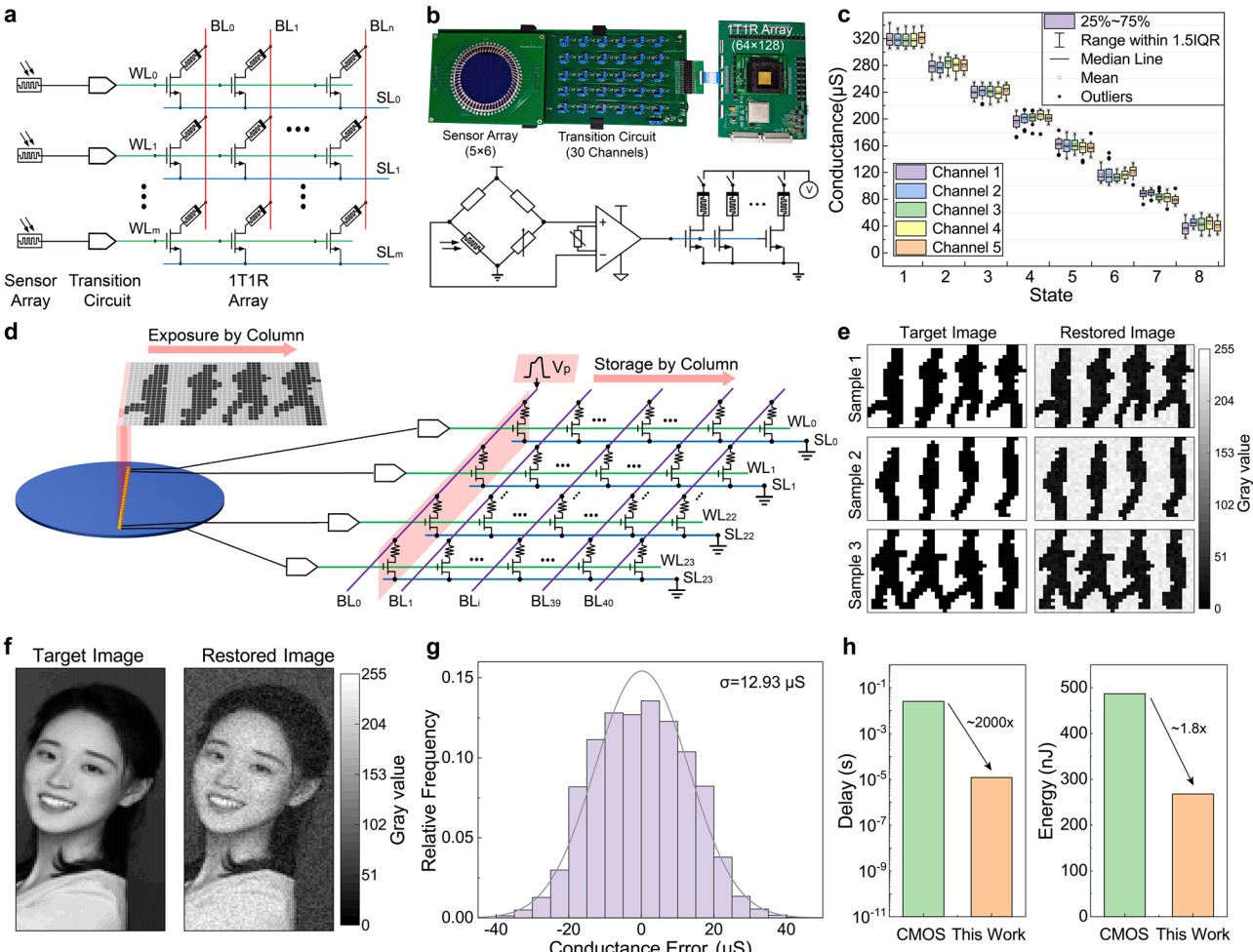

**Fig. 3 | Schematic diagram of the 1PnR architecture and the rolling exposure strategy for fast image acquisition. a** Each pixel sensor is connected to a gate line of the 1T1R array, constructing a one-pixel-multiple-memristor architecture for sequential image acquisition. **b** The scheme and electric board of the transmission circuit, where the pixel's photo-resistive signal is converted to gate voltage by a differential amplifier. **c** Distribution of the device conductance on the electric board using the OPM writing strategy. Five devices on different channels are successfully modulated to eight states. **d** The rolling exposure strategy starts from the first column of the image using the OPM method, then rolls down one by one, until

the last column. **e** Experimental result using rolling exposure strategy for three samples of the Weizmann dataset, left is the target image, while right is the restored image read from the memristor array. **f** Experimental results of the portrait image using rolling exposure strategy, the facial features in the restored image are still clear for classification with negligible noise. **g** Distribution of the sensing error of the restored image in (**e**), which is consistent with the OPM performance with a normal distribution. **h** Comparison of delay and power consumption between the proposed 1PnR architecture and a typical digital system composed of CMOS sensors and DDR5 memory.

---

tends to be uniform under pulse amplitude. The 8 gate voltages ranging from 1.21 V to 2.0 V, leading the corresponding conductance averages in 40–320 μS, with an interval of 40 μS. Figure 2f shows the distribution of the eight conductance states in the form of a box plot. The rectangular box represents the 25–75% distribution range, and the upper and lower edges represent 1.5 times the interquartile range. All the device conductance fluctuates near the target value, except for some outliers. And the overall fluctuation range is within 20 μS (half of the interval). It should be noted that the pulse waveform is designed with two voltage segments (1.2 V and 1.6 V) to prevent impact responses caused by voltage jumps from affecting the modulation stability. The experiments of a 1-segment pulse are also performed and plotted in Supplementary Fig. 3, and the modulation error of device conductance is compared in Fig. 2g. The results reveal that the 2-segment pulse shows narrow distribution in modulation error, which has better stability. Furthermore, retention tests of the eight states are performed, and the result plotted in Supplementary Fig. 4 shows that the device states can last up to 50,000 s, with only small fluctuation.

These results demonstrate the great compatibility of the 1T1R cell with the OPM method.

## Rolling exposure on a one-pixel multiple-memristor structure

In this work, an architecture integrating the pixel sensors and memristor array is proposed for a neuromorphic near-sensor computing system, which we call the one-pixel-multiple-memristor (1PnR) Structure. The schematic diagram of the 1PnR structure is shown in Fig. 3a, where pixel sensors are connected to the gate lines (WL) of the 1T1R array one by one through analog conversion circuits. The circuit has a very simple structure, containing a voltage division part and a differential amplifier, which can convert the light-induced resistance change of the pixel sensor to voltage values. To sufficiently verify the 1PnR structure, 30 channels of image sensors and conversion circuits are experimentally demonstrated, as shown in Fig. 3b. The circuit parameter is adjusted to fit the gate voltage range of the 1T1R array, and the transmission relationship of the circuit from light density to gate voltage, as well as the transmission speed is plotted in Supplementary

Fig. 5. The 8k memristor chip is deployed on a transfer board, where all the pins can be controlled by the 1PnR system to apply OPM pulse. Then, pixel stimuli with eight different intensities are applied to five randomly picked channels to verify the sensing-storage ability of the 1PnR board. The light intensities are experimentally picked to achieve the same gate voltages in Fig. 2f for comparison, and 50 repeated tests are performed under each light intensity. Figure 3c shows the distribution of memristor conductance after the experiments. The statistical results are presented in box plots, with different devices distinguished by colors. As can be seen from the figure, the modulated conductance fluctuates around the target value, and the distribution maintains high consistency between channels. Compared with the results in Fig. 2f, the 1T1R cells in 1PnR achieved the same eight-state modulation under the same parameters, and the distribution range was also roughly the same. The results indicate that the conversion circuit makes considerable compatibility with the pixel sensors and 1T1R cell, which can support the 1PnR structure to achieve fast storage of pixel images.

Based on the 1PnR architecture and OPM method, a rolling exposure strategy (RES) is then proposed for fast image acquisition. The workflow of RES is shown in Fig. 2d, taking the Weizmann human action dataset[33] as an example, where the image is binarized and cropped to $24 \times 40$. On the left side is a column of 24-pixel sensors grown on a silicon wafer, which are connected to the gate lines of a $24 \times 40$ 1T1R array through conversion circuits, establishing 24 channels of 1PnR structure. Before image exposure, the memristor array is initialized to the same HRS, and the SLs are all grounded. When the first column of the pixel image is under-exposed, the pixel information can be converted to gate voltages by the conversion circuits at the same time. Then, the OPM is applied to the first column ($RL_0$) to store the pixel information in parallel. Next, the same exposure method is used to store the image in the memristor array column by column, until the entire image is exposed and stored.

Utilizing the fabricated pixel sensors and transmission circuit, a 1PnR hardware verification system is implemented, as shown in Supplementary Fig. 6. UV LED light sources are employed, corresponding one-to-one with the fabricated pixel sensors. The image is converted into UV light signals based on pixel values and directly irradiated to the optical sensors, which simulates the optical exposure process. And the detailed process of the exposure experiment based on the hardware system is illustrated in Supplementary Fig. 7. Besides, to achieve RES verification with a larger image size based on the prepared image acquisition hardware, the target column of the image (up to 30 pixels) is reshaped to a $5 \times 6$ array for UV irradiation. The light signals received by the optical sensors are then reshaped back to one column through the 30-channel conversion circuits and then written to the target column of the connected memristor array, which is shown in Supplementary Fig. 8.

RES experimental verification is firstly conducted using three sample images picked in the Weizmann dataset, containing actions of run, jump and walk. Each sample is processed into a $24 \times 40$-pixel image, formed by concatenating 4 consecutive $24 \times 10$-pixel frames of human motion, as illustrated in Supplementary Fig. 12. During the image acquisition experiment of each sample image, 24 sensing channels are used according to the column pixels, and a total of 40 rolling exposure steps are required for the whole sample image. After the RES experiment, the target image and restored image from the memristor array are plotted in Fig. 3e. It can be seen from the results that the action features in the restored images are still clear for classification with negligible noise.

To further verify the RES strategy, a hardware experiment utilizing a gray-scaled image and a whole 8k memristor array is performed. A portrait image is chosen as an example, which is gray-scaled and cropped to a $128 \times 64$ scale to match the memristor array. According to the design of RES, if the 128 gate lines of the 8k array are fully utilized, a $128 \times 64$ image can be perceived in just 64 exposure operations. However, to match the 30-channel circuit prepared in the experiment, the image is divided into 5 groups with 30 pixels, as shown in Supplementary Fig. 9. The rolling exposure of $30 \times 64$ is completed within each group first, and then the next group of images is exposed in sequence. Fig. 3f shows the target image and restored image from the memristor array after the RES experiment. Although the restored image has pixel noise caused by writing errors, the original facial features are still clearly distinguishable, which is sufficient to support applications such as face recognition. Figure 3g shows the distribution of the writing error of the portrait image, which is consistent with the OPM experiment with a normal distribution. Indeed, the standard deviation becomes slightly larger due to the conversion of gray pixel values. Fig. 3h shows the performance comparison between the 1PnR architecture with a typical CMOS system consisting of image sensors and DDR5 memory. The comparison details are introduced in Supplementary Note 1. The results reveal that the proposed 1PnR system has ~2000 times reduction of time latency and 1.8 times reduction of energy consumption, which further demonstrates the great potential in fast image acquisition applications.

In fact, the proposed 1PnR architecture exhibits remarkable flexibility and scalability for image acquisition. For future optical system applications, when the sensor array scale becomes sufficiently large, it can function as an area sensor for global exposure. Following the operational principle of line-scan CCD cameras, the entire image can first be exposed globally, and then its pixels can be transferred column by column to the memristor array for storage. If multiple memristor arrays are employed, each column of sensors can be connected to a different memristor array. After exposure, writing pulses can be applied simultaneously to different arrays, enabling the entire image to be written into separate memristor arrays at the same time, offering very fast image exposure, as illustrated in Supplementary Fig. 10. Moreover, if the scale of the memristor array is sufficiently large, all sensor units in the area array can be connected to the same array, allowing an entire image to be written into a single column of memristors.

## Data-in-situ computing network for sequential vision

In a traditional memristor-based neuromorphic system, the network weights are mapped to the conductance values of the memristor array, and the samples are mapped to voltage signals by the artificial neurons. The memristor array can achieve the MAC operation efficiently through Ohm's law and Kirchhoff's current law in the analog domain. However, in a near-sensor integrated computing system, the image data has already been stored in the memristor array. Performing calculations in the traditional way requires the image data to be transferred from the storage array to the computing array, resulting in a large amount of transmitted data, which reduces the overall efficiency of the computing system. Here, we utilize the MAC feature of the data storage array and propose a data-in-situ computing network for memristor-based neuromorphic systems. As shown in Fig. 4a, the first part of the network weights is mapped to the voltage weight vector (VWV), which is applied to the data-storage memristor for data-in-situ computing. Then the computing results are fed into a perception for classification, which is implemented on a weight-storage memristor array. The data-in-situ computing weights are updated through back-propagation calculation, along with the perception weights in the training process. Utilizing a data-in-situ network, the image is processed where it is stored, only the data-in-situ computing results with greatly decreased data amount are required for transmission for classification, which is consistent with the high-efficiency mechanism of the visual working memory in the human brain.

To evaluate the data-in-situ computing network for sequential image processing based on the 1PnR system, the Weizmann Dataset[33] is used for evaluation, which has 10 actions, containing bend, jack, jump,

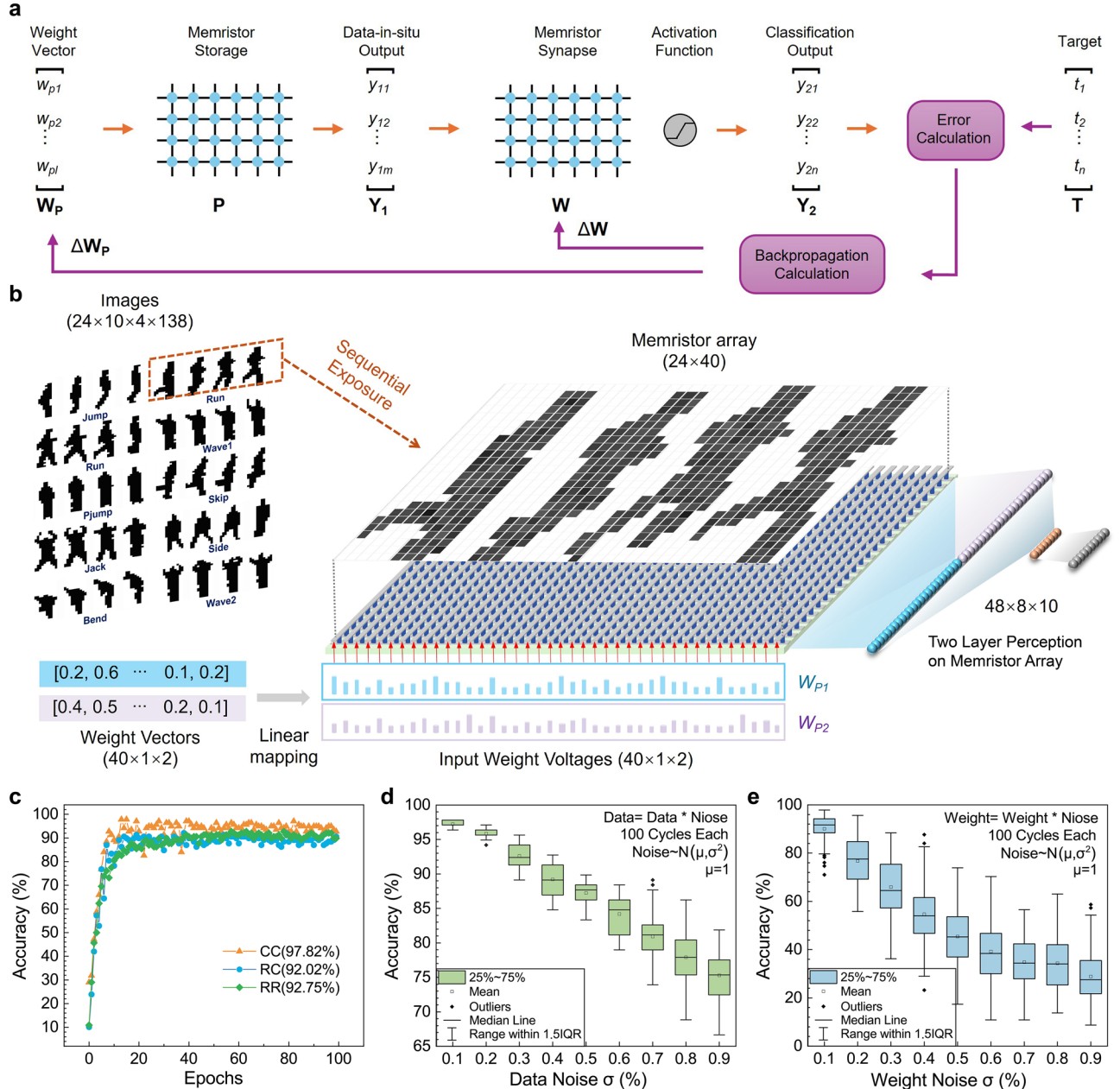

**Fig. 4 | Data-in-situ computing network for image processing. a** Diagram of the data-in-situ computing network architecture. The voltage vector carrying network weights is applied to the data-storage memristor for data-in-situ computing, then the results are fed into a perception for classification. The data-in-situ weights are updated through backpropagation calculation, along with the perception weights in the training process. **b** The schematic diagram of the data-in-situ computing network for the Weizmann human action dataset based on a memristor array. On the left are examples of the dataset. Two voltage vectors carrying the network weights are applied to the array for data-in-situ computing, and the results are then fed to a two-layer perception realized on a memristor array for classification. **c**–**e** recognition results for the Weizmann dataset of the network simulation, the input voltage directions, image noises and input weights noises are analyzed, respectively.

pjump (jump in place), run, side, skip, walk, wave1 (in one hand) and wave2 (in two hands). Each action is performed by 9 people. The dataset is already binarized and aligned from video clips. In our simulation, the dataset is cropped to 24 × 10. To make sure that every sample includes a whole period of periodic actions (such as walk, run and wave), 4 frames are averagely picked from 12 consecutive frame sequences in the video clip to construct one sample, as shown in Supplementary Fig. 12. Furthermore, a leave-one-out strategy is used in the sample preparation, which is, 8 people are chosen for training (1065 samples for 10 actions) and the rest 1 person is used for testing (138 samples for 10 actions).

The network structure is illustrated in Fig.4b, where one sample is stored in the 24 × 40 memristor array by RES. For the Weizmann dataset classification, two VWVs (40 × 1 × 2) are applied from the column of the array for data-in-situ computing successively, and the results are joined together (48 × 1) and then fed into a 48 × 8 × 10 two-layer perception for image classification. The data-in-situ computing network is first verified by simulation, 97.82% accuracy can be obtained for the dataset after 100 training epochs, showing great efficiency of the network structure. Furthermore, the impact of VWV direction for data-in-situ computing is analyzed. As shown in Fig. 4c, when 2 VWVs are both applied from the column of the image, the classification

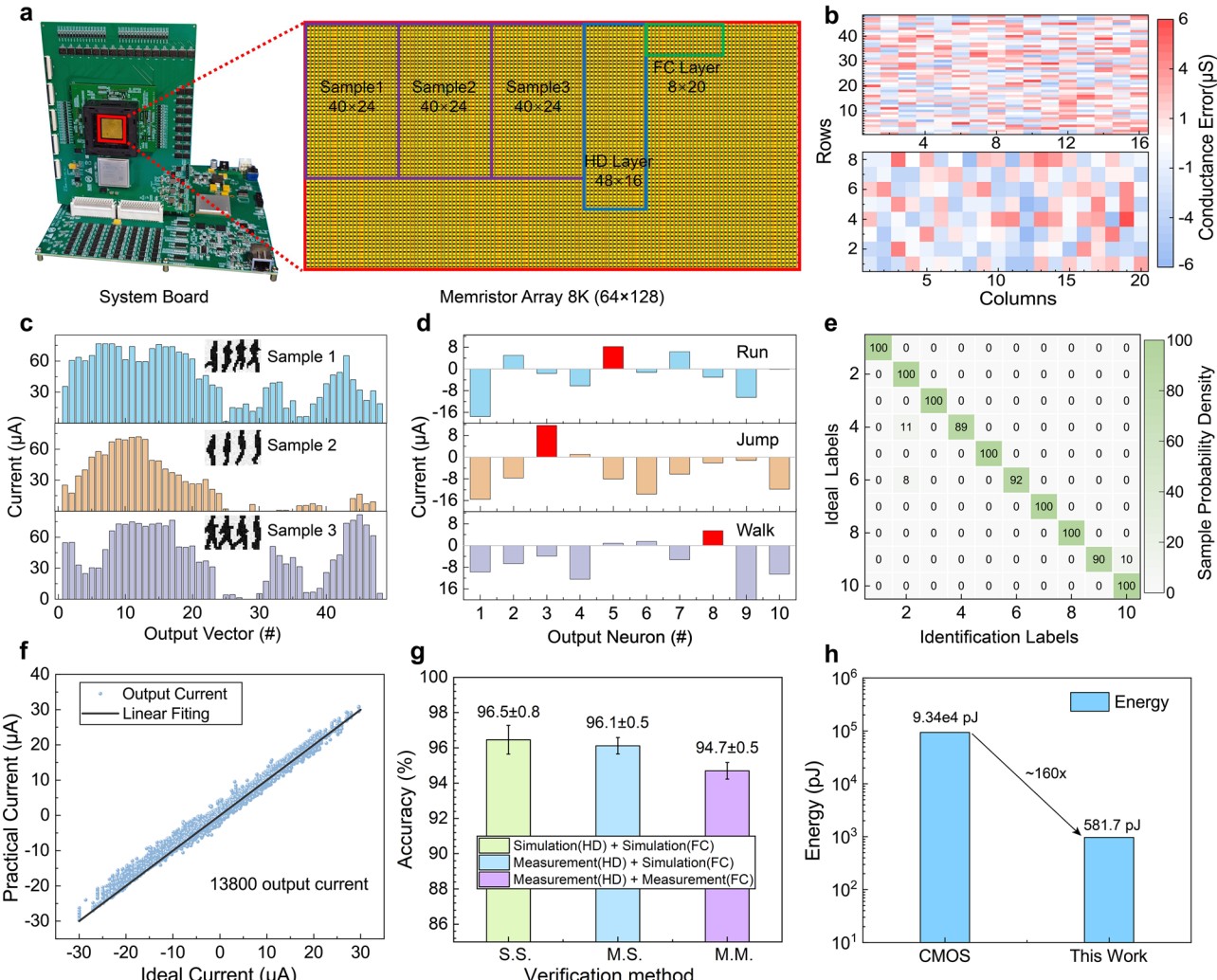

**Fig. 5 | Experimental demonstration of the 1PnR hardware system. a** Optical image of the memristive hardware system, where the memristor array is enlarged. The color blocks in the array reveal the partitions to implement the data-in-situ computing network for Weizmann classification. **b** Weight-transfer errors of the HD (48 × 8) and FC (8 × 10) layers to the memristor array by differential group. The transfer errors are limited within 6 µS by an array modulation script. **c**, **d** Data-in-situ computing current and perception computing current from the hardware system for the three test samples. **e** Recognition results for the Weizmann dataset of the hardware system, with an accuracy of 95.7%. The accuracy decay (2.1%) compared to the ideal values may be caused by the mapping error of the network weight. **f** Comparison between the ideal calculation current and the hardware output current of all the test samples. **g** Classification results of both simulations in the HD and FC layers of perception, experiments in the HD layer and simulation in the FC layer, both experiments in the FC layer. The error bars represent the mean value and standard deviation of network accuracy obtained from ten repeated experiments with random noise. **h** Estimated energy consumption of the data-in-situ computing architecture and a typical CMOS system for Weizmann classification, a ~160 times reduction can be achieved.

accuracy reaches best to 97.82%, if 2 VWVs are both applied from rows of the array, the accuracy reduces to 92.75%. And if 1 VWV from column and 1 from row, the accuracy is 92.02%. It can be conducted that the sample can keep more features after data-in-situ computing in the column direction and achieve better classification performance. Finally, the impact of image noise and weight noise on the network is simulated, which are plotted in Fig. 4d, e, respectively. The noise percentage is applied in a randomly generated normal distribution, with a mean value of 1. Each standard deviation is simulated for 100 cycles. The results show that as the noise deviation rises, the network accuracy decreases accordingly. But the noise on image data shows less impact on the performance, indicating better robustness in the image sensory error.

Finally, Experimental verification of the data-in-situ computing network system is performed on the 1PnR hardware system. Fig. 5a shows the optical photo of the memristor modulation board. With the support of the core controller (FPGA) and the array read-write circuit

(ADC&DAC), memristor modulation and the MAC computation can be achieved by the system. The mapping layout of the data-in-situ computing network in the memristor array is enlarged. Columns 0–71 are assigned for sample image storage and data-in-situ computing, which can store 3 samples, as shown in the purple box. The hidden layer (HD, 48 × 8) of the perception needs 16 columns after differential process and is mapped to columns 72–87 of the array. Besides, the fully connected layer (FC, 8 × 10) requires 20 columns, which is mapped to the final 88–127 columns. The network weights (HD and FC) are mapped to the memristor array in the range of 100–200 µS by a modulation script at the host computer. The script adopts a closed-loop modulation strategy, and the maximum conductance error is limited to 6 µS to achieve a balance between weight modulation speed and network accuracy. Fig. 5b shows the results of the HD and FC layer mapped to the memristor array, which presents 100% yield for the network computation.

During the experiment, every three samples are stored to the assigned area through OPM by experiments shown in Fig. 5a. Then the data-in-situ computing of the image is performed by the hardware system, and the output currents of the 3 representative samples are plotted in Fig. 5c. The results are obviously different between the three samples, which indicates that data-in-situ computing can successfully extract the sample features for subsequent classification. Next, the results are fed into the HD and FC layers on the hardware successively. The output current of the FC layer for the three representative samples is plotted in Fig. 5d. The largest current is marked in red, representing the classification result. It is clear that the three samples are both successfully classified. Then, all the test samples are applied to the system for hardware verification. A typical recognition result of the hardware system is shown in Fig. 5e, with an accuracy of 95.7%. The 2.1% decay compared to the simulation results may be caused by the mapping error of the network weight. The comparison between the ideal calculation current and the hardware output current of all the test samples in the FC layer is plotted in Fig. 5f, which shows considerable consistency.

To further evaluate the system performance and stability, repeated experiments are performed combining simulation and hardware measurements. During simulations, random conductance errors in the image acquisition process with a normal distribution are applied, which is consistent with a practical experiment. The results from both simulations in the HD and FC layers, experiments in the HD layer and simulations in the FC layer, and both experiments in the FC layer are presented in Fig. 5g; each combination is repeatedly performed for 10 times. The network accuracy with both simulations in two layers reaches $96.5 \pm 0.5$ (mean value 96.5 with standard deviation 0.5) and slightly decreases to $96.1 \pm 0.5$ when using experimental results in the HD layer and simulation in the FC layer. The accuracy loss in experimental results may be caused by the random electric noise in the hardware system. At last, the result of both hardware experiments in two layers reaches $94.7 \pm 0.5$, demonstrating the good robustness of the hardware system.

Furthermore, the energy consumption of the data-in-situ computing architecture for the Weizmann Dataset is estimated; details are introduced in Supplementary Note 2. A CMOS system consisting of a typical digital accelerator-based (HNPU-based) system[34] with DDR5 memory[35] is estimated for comparison, which is plotted in Fig. 5f. The results show that the proposed data-in-situ computing architecture has about 160 times the energy efficiency of typical CMOS systems.

To evaluate the general classification ability of the data-in-situ computing network, the benchmark is then performed on the commonly used MNIST[36] and Fashion MNIST[37] datasets with noise analysis. The experimental results on the MNIST dataset are presented in Supplementary Fig. 13 and Supplementary Note 3, which demonstrate a 95.9% (simulation) and 92.4% (experimental) classification accuracy. The simulation results on the Fashion MNIST dataset are shown in Supplementary Fig. 14. Given the greater complexity of Fashion MNIST images compared to MNIST, 4 voltage vectors are applied to the image for data-in-situ computing to extract more image features, which achieves an 87.17% recognition accuracy. Finally, the comparison of the 1PnR system and other representative works about the integrated sensing-storage-computation system is performed and summarized in Supplementary Note 4. These results further demonstrate the great performance of the data-in-situ computing network for future neuromorphic visual systems.

## Discussion

In this work, a one-pixel-multiple-memristor (1PnR) computing architecture is proposed for sequential image sensing, storing and processing. The sensors and memristor array are integrated by a gate multiplexing architecture, offering the ability to store a single channel of optic input to multiple memristor cells simultaneously. Then, a rolling exposure strategy for fast sequential image acquisition is proposed. With the pixel image sensed and stored column by column to the memristor array in the analog domain, both time and energy efficiency can be achieved compared to a traditional CMOS system. Moreover, a data-in-situ computing network based on a memristor array is proposed for fast image processing. Voltage vectors carrying network weights are applied to the image storage array to implement data-in-situ computing, which avoids transmission of the image data in traditional memristive neuromorphic systems. Finally, a 1PnR hardware system is established for verification of sequential visual sensing and processing. The experimental results reveal 95.7% recognition accuracy for the Weizmann human action flow dataset, demonstrating the great classification capacity of the data-in-situ network. Compared to typical CMOS-based systems, the proposed architecture is estimated to have a 2000 times reduction of time latency for image sensing and storage, and 160 times reduction of energy consumption for image processing, demonstrating the great potential for constructing neuromorphic visual systems.

## Methods
### Sample fabrication
The fabrication process of the ITO/ZnS/TiN sensor. The bottom electrode (TiN, ~40 nm) is deposited by magnetron sputtering, then it is patterned by lithography and wet etching. After the second lithography process, the functional material zinc Sulfide (ZnS, ~30 nm) and transparent top electrode ITO (~30 nm) are deposited by magnetron sputtering, respectively. The effective device size is determined by the round area in the bottom electrode, with a diameter of 200 μm.

Fabrication of an 8k memristor chip. The chip has a $64 \times 128$ one-transistor-one-resistor (1T1R) memristor array. The transistors in the array are fabricated using a standard 180 nm Si complementary metal−oxide semiconductor process. The memristor cell has a structure of $TiN/TaO_x/HfO_x/TiN$. 12 nm $HfO_x$ is deposited by atomic layer deposition as a resistive switching material, then 45 nm $TaO_x$ is deposited by magnetron sputtering as a thermal enhanced layer. The TiN electrode is deposited by magnetron sputtering.

### Measurement method
The electrical test of a single device is performed with the Keithley 4200 SCS semiconductor parameter analyzer. The optical light is provided by UV LEDs with a wavelength of 365 nm, and a signal generator is used to control the UV light pulse. The memristor chip and network calculation are measured by our memristor-evaluation system, which is equipped with a core controller (FPGA) and array read-write circuits (ADC&DAC).

### Network simulation method
The network simulation is conducted using a Python environment and the PyTorch framework. The image samples are from the Weizmann Dataset of human action flows, which are preprocessed to $24 \times 40$ binary images. During the training process, two sets of weight vectors applied to the samples are treated as updatable parameters and iteratively updated using the backpropagation algorithm. The ReLU activation function is employed, and the cross-entropy loss function is used for error computation.

### Research participant consent
The authors affirm that human research participants provided informed consent for publication of the images in Fig. 3f, Supplementary Fig. 8 and Supplementary Fig. 9.

### Data availability
The source data generated in this study are provided in the Source Data file. Additional data related to this paper can be requested from the authors. Source data are provided with this paper.

## Code availability

The code of the data-in-situ computing simulation will be available from the corresponding authors upon request.

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

## Acknowledgements

This work was supported by the National Key R&D Program of China under Grant Nos. 2024YFA1208800 (Q.L.) and 2023YFB4502200 (Y.Y.), Innovation Research Foundation of National University of Defense Technology under Grant Nos. ZK24-09 (Y.S.) and 25-ZZCX-JDZ-18 (W.W.), National Natural Science Foundation of China under Grant Nos. 62404253 (Y.S.), 62304254 (W.W.) and U23A20322 (Q.L.), Guangdong Provincial Key Laboratory of In-Memory Computing Chips under Grant No. 2024B1212020002 (Y.Y.), Shenzhen Science and Technology Program under Grant No. JCYJ20241202125907011 (Y.Y.), as well as the Beijing Natural Science Foundation under Grant Nos. L234026 (Y.Y.) and L257010 (Y.Y.).

## Author contributions

Y.S., P.W.T., and W.W. designed the experiments. Y.S. designed and fabricated the sample devices and conducted device-level electrical experiments. C.L.C. and Y.N.W. designed and fabricated the circuit boards. J.R.S. designed and simulated the task algorithms. Y.S. and P.W.T. designed a data-in-situ computing architecture. H.X., R.R.C., C.L., and B.S. assisted with data analysis and interpretation. Y.S. and P.W.T. co-wrote the manuscript. All authors discussed the results and revised the manuscript. W.W., Y.C.Y. and Q.J.L. supervised the research.

## Competing interests

The authors declare no competing interests.
