## [Transparent Peer Review file · Nature Communications]

Data-In-situ Computing with One-Pixel-Multiple-Memristor Architecture for Neuromorphic Sequential Vision

Corresponding Author: Professor Wei Wang

Version 1:

Reviewer comments:

Reviewer #1

(Remarks to the Author)

This manuscript proposes a novel neuromorphic vision system based on a one-pixel-multiple-memristor (1PnR) architecture and demonstrates significant reductions in latency (~2000×) and energy consumption (~160×) compared to CMOS systems. While the system-level integration and experimental validation of data-in-situ computing are commendable, the imaging demonstration is oversimplified and lacks sufficient detail. Specifically, the imaging setup is not described rigorously, and the pathway from optical sensing to meaningful image reconstruction is ambiguous. These omissions undermine the credibility of the proposed visual acquisition scheme.

The manuscript claims sequential image acquisition using a 5×6 optical sensor array (Supplementary Fig. 1a), but it provides no description of how the object is positioned, how light is collected, or whether a lens system is employed. Without this, the nature of the "captured image" is unclear and the spatial correspondence is speculative.

Figure 3e and 3f present restored images (Weizmann silhouettes and a human portrait) that show far greater resolution and image alignment than the 5×6 sensor array could possibly provide. This implies either strong preprocessing or assumptions not disclosed in the paper.

The mechanism by which the sensed data from the ITO/ZnS/TiN pixel sensors is relayed to the memristor array via the transition circuit (shown in Fig. 3a) lacks quantitative or temporal detail. What is the timing resolution, and how is alignment ensured during the rolling exposure process?

For Figure 3f (portrait image), Supplementary Figure 5 explains a division into 5 blocks of 30×64 segments, but again, there is no mention of the optical projection method, scene conditions, or how misalignment or motion was controlled. This raises concerns that the "image" may be more synthetic or pattern-based than optically formed.

The claim that a 24×40 image from the Weizmann dataset can be reconstructed column by column assumes each column corresponds directly to an optical sensor signal. However, the paper doesn't clarify whether a lens forms a spatial image or if each sensor merely encodes a temporal sequence. Without this, the legitimacy of the image modality is uncertain.

The object-imaging chain — from real-world object, through optical transduction, to data conversion and final classification — is not sufficiently detailed to confirm the system's operation as a real imaging device rather than a neuromorphic signal processor using artificially prestructured inputs.

(Remarks on code availability)

Since I am a hardware engineer, I am not in the best position to evaluate the provided source code. However, I recommend referring to my review regarding the measurements and data interpretation.

Reviewer #2

(Remarks to the Author)

This paper introduces a one-pixel-multiple-memristor(1PnR) architecture to enhance image processing efficiency. By connecting each pixel to multiple memristors, the authors enable parallel storage and in-situ computing within a memristor

array, significantly reducing data transmission overhead. The proposed system incorporated a rolling exposure strategy for high-speed image acquisition and reported 95.7% accuracy on the Weizmann human action dataset. Additionally, the system showed improvements in latency and energy consumption compared to conventional CMOS-based systems. However, despite the practical demonstration, this manuscript suffers from insufficient explanation.

Details are elaborated below:

1) The claimed benefits of the 1PnR architecture over previously reported sensor-memory-compute integrated systems are not clearly demonstrated. Quantitative or qualitative comparisons are necessary to justify the advantage of this architecture beyond the reported speed and power gains.

2) The manuscript lacks a detailed discussion on how the proposed architecture could be scaled or implemented as fully integrated system.

With respect to the concerns outlined above, the following issues should be addressed:

Comment #1:

In 'Introduction', The manuscript provides a compelling vision for neuromorphic visual processing; however, the novelty of the proposed 1PnR architecture in comparison to existing sensor-memory-compute integrated systems—such as 2D/3D-stacked neuromorphic imagers or hybrid CMOS-memristor solutions—is not sufficiently highlighted. Including a more explicit architectural or performance comparison with recent state-of-the-art systems would help better situate this work within the current research landscape and substantiate its unique contributions.

Comment #2:

In 'Data-in-situ computing network for neuromorphic sequential vision' section, While the use of the Weizmann human action flow dataset demonstrates the classification capability of the proposed system, it would be beneficial to evaluate the model on more complex and diverse datasets. In particular, testing under real-world variations such as lighting changes, occlusions, and sensor noise could better establish the robustness and generalizability of the 1PnR architecture.

Comment #3:

The manuscript presents a promising hardware implementation of the 1PnR system. However, it remains unclear whether the entire system—including the optical sensor, analog front-end, memristor array, and in-situ computation unit—has been physically fabricated and tested as an integrated platform. A detailed description of the experimental setup, along with discussions on system-level limitations such as scalability, endurance, and noise resilience of the memristor devices, would significantly strengthen the claims on practical viability.

Comment #4:

Although the architecture is described as bio-inspired, the biological underpinnings—particularly the function and mechanism of visual working memory in the human visual system—are only briefly introduced. Providing a more thorough explanation of how specific components of the proposed system (e.g., sensor, memory, and compute modules) correspond to biological elements such as the retina, visual cortex, and working memory would enhance the conceptual grounding and clarify the biomimetic relevance.

Comment #5:

Despite the authors performed the classification of binary(black and white) images, there is limited discussion regarding the handling of color images. Given that the memristor-based system supports multiple resistance state, it would be valuable to evaluate whether this architecture can maintain performance when dealing with color images. Further investigation into this aspect would strengthen the paper's applicability to a broader range of real-world image processing task.

(Remarks on code availability)

Reviewer #3

(Remarks to the Author)

This manuscript, "Data-In-situ Computing with One-Pixel-Multiple-Memristor Architecture for 1 Neuromorphic Sequential Vision," is inspired by the working memory mechanism of the human visual system and proposes a novel neuromorphic visual architecture with a single-pixel multiple-memristor (1PnR) structure. Additionally, the authors propose a rolling exposure strategy based on open-loop single-pulse modulation, which leverages the array architecture of the 1PnR to enable column-by-column sensing of images while simultaneously storing signals. This strategy enhances the system's temporal and energy efficiency. Finally, the authors propose a resistive-based in-situ data computation network distinct from traditional resistive networks, mapping voltage signals as weights and conductance as image data. This method eliminates the need for physical transmission of image data, instead performing computations directly, significantly improving system efficiency. Finally, the authors constructed a 1PnR hardware system comprising 30 optical sensors and an 8k29 memristor array, achieving an image recognition rate of 95.27% on the Weizmann Human Action Flow Dataset. This work is highly innovative, with the proposed strategies and algorithms effectively addressing issues faced by traditional methods, and holds significant value in applications such as image recognition, processing, and analysis. However, there are still some problems that need to be solved in this work.

(1) There are two errors in the image descriptions. Please confirm and correct them if they are indeed errors: First, the reference to Figure 2f in line 160 of the manuscript should refer to Figure 2g; second, the reference to Figure 2g in line 234 of the manuscript should refer to Figure 3g.

(2) There is a problem with the subscript in image 2d. Please correct it.

(3)The fonts in the image are inconsistent. Please correct them. In addition, the image is not clear enough. Please replace it with a high-definition image.

(4)The manuscript only shows optical images of the array and TEM images of the memristors. Please add a diagram of the device structure of a single unit in the array (a schematic diagram of the structure connecting the transistor and the memristor).

(5)The article presents the electrical characteristics of memristors and individual units, but does not present the electrical characteristics of transistors. Please supplement the electrical characteristics of transistors in the array, such as transfer characteristic curves and output curves, to further prove the principle of array function implementation.

(6)Supplemental Figure 1, the unit of light intensity in the 17th row of the image description does not match that in Supplemental Figure 1b.

(7)The supplementary figures 1b and 1c show changes in voltage as well as switching of light conditions, which do not clearly demonstrate the performance of the light sensing unit. Please supplement the relevant measurements to prove the performance of the light sensor more rigorously.

(8)Please explain the RS window in detail.

(Remarks on code availability)

Version 2:

Reviewer comments:

Reviewer #1

(Remarks to the Author)

The revised manuscript is well organized, and ready for publication.

(Remarks on code availability)

Reviewer #2

(Remarks to the Author)

Thank you for the authors' diligent revisions. The responses adequately address the prior concerns, and I support publication of this manuscript in Nature Communications.

(Remarks on code availability)

Reviewer #3

(Remarks to the Author)

The issues raised by the reviewers have been addressed. I believe the manuscript can be published in its present form.

(Remarks on code availability)

Reviewer #1:

This manuscript proposes a novel neuromorphic vision system based on a one-pixel-multiple-memristor (1PnR) architecture and demonstrates significant reductions in latency ($\sim 2000\times$) and energy consumption ($\sim 160\times$) compared to CMOS systems. While the system-level integration and experimental validation of data-in-situ computing are commendable, the imaging demonstration is oversimplified and lacks sufficient detail. Specifically, the imaging setup is not described rigorously, and the pathway from optical sensing to meaningful image reconstruction is ambiguous. These omissions undermine the credibility of the proposed visual acquisition scheme.

Comment 1:

Reviewer wrote:

The manuscript claims sequential image acquisition using a 5×6 optical sensor array (Supplementary Fig. 1a), but it provides no description of how the object is positioned, how light is collected, or whether a lens system is employed. Without this, the nature of the "captured image" is unclear and the spatial correspondence is speculative.

Our response:

Thank you for your review comments. We only provide a brief description of the image acquisition experiment in the initial version of the manuscript, and we apologize for any misunderstanding. Here, we will elaborate in detail on the design and principles of the image acquisition experiment.

It should be noted that our image acquisition experiment is a proof-of-concept experiment, primarily designed to demonstrate and validate the proposed 1PnR architecture and the rolling exposure strategy. No lens system is used to expose actual objects. Instead, we combine UV LED light sources into an array, corresponding one-to-one with the fabricated optical sensor array. The image is converted into UV light signals based on pixel values and directly irradiated onto the optical sensors to simulate the optical exposure process. This experiment setup allows for easy switching between different image samples for recognition and avoids the influence of lens imaging quality on system, which enables us to focus more on the validation of the 1PnR architecture and the data-in-situ computing network, rather than the implementation of a complete optical imaging chain. The specific details of the experiment are as follows:

Figure R1. Hardware implementation of the image acquisition experiment. The UV LED array board and UV sensor array board are positioned and fixed by one-to-one for light emitting and sensing per pixel. Opaque grids are

applied to avoid cross-talk between light signals of different pixels. The 5×6 optical sensors are then connected to 30 channels of transmission circuit, which converts light information to gate voltages of memristor array for further image storage.

1. Hardware Implementation of the Image Acquisition Experiment

The hardware part of the image acquisition experiment is illustrated in Figure R1. A 5×6 optical sensor array is fabricated on a 4-inch silicon wafer, and a 5×6 UV LED array board is designed corresponding one-to-one with the optical sensors, enabling pixel-by-pixel light signal transmission and sensing. Additionally, 3D-printed light-opaque grids are applied to isolate different LED light sources and optical sensors, preventing cross-talk between light signals of different pixels. The LED light source board and the optical sensor array board are precisely aligned and fixed using 3D-printed interlocking parts.

The optical sensors are not interconnected in the array. The electrodes of each sensor are connected via copper wires to the PCB baseboard and then linked to the gates of the memristor array through conversion circuits. Corresponding to the number of optical sensors, the board contains 30 channels of conversion circuits, which can convert the light information simultaneously to gate voltages of the memristor array for further image storage.

The UV LEDs are driven by power MOS chips. During the experiment, the gate voltage of the power MOS is controlled according to the target pixel grayscale value to produce the corresponding LED light intensity. To generate 30 channels of gate voltage control signals, we also developed an LED control circuit board. The MCU on this board can receive instructions from the PC host computer and control multiple DAC chips to generate 30 channels of gate voltage control signals. The conversion relationship between pixel grayscale values and the gate voltage of the power MOS is pre-calibrated based on overall system experiments.

Figure R2. The Schematic diagram of light collection mechanism for rolling exposure strategy. Each column of the target image is reshaped to 5×6 array for exposure experiment. The received signals are reshaped back to one column through the 30-channel conversion circuits and further written to the corresponding column of memristor array in parallel using the proposed One-Pulse Method. Then, next column of image is exposed and stored to the corresponding next column, till the whole image is sensed and stored.

2. Light Collection Mechanism for Image Rolling Exposure

In the manuscript, the proposed rolling exposure strategy senses and stores the image column by column. To achieve experimental verification with larger image size based on the prepared image acquisition hardware, each

column of the image (30 pixels) is reshaped to a 5×6 array and is applied to the optical sensor array via the UV LED array. The light signals received by the optical sensors are then reshaped back to one column through the 30-channel conversion circuits and connected to the gates (WLs) of the 1T1R memristor array. The image column data is further stored in the corresponding column of the memristor array in parallel using the proposed One-Pulse Method. Then, next column of image is exposed and stored to the corresponding next column, till the whole image is sensed and stored. The entire process is shown in Figure R2. This experimental approach allows us to achieve verification of rolling exposure strategy with larger image size as effectively as possible.

We appreciate the opportunity to elaborate on these details and hope that this explanation enhances the clarity and credibility of our visual acquisition scheme. To clarify this in the revise manuscript, we have added a detailed introduction of the 1PnR hardware implementation, and **modified the manuscript (Section 2.2, Paragraph 3,4, Highlighted in yellow)** as:

Utilizing the fabricated pixel sensors and transmission circuit, a 1PnR hardware verification system is implemented, as shown in **Supplementary Fig. S6**. UV LED light sources are employed, corresponding one-to-one with the fabricated pixel sensors. The image is converted into UV light signals based on pixel values and directly irradiated to the optical sensors, which simulates the optical exposure process. And the detailed process of the exposure experiment based on the hardware system is illustrated in **Supplementary Fig. S7**. Besides, to achieve RES verification with larger image size based on the prepared image acquisition hardware, the target column of image (up to 30 pixels) is reshaped to a 5×6 array for UV irradiation. The light signals received by the optical sensors are then reshaped back to one column through the 30-channel conversion circuits and then written to target column of the connected memristor array, which is shown in **Supplementary Fig. S8**.

Correspondingly, we have modified the Supplementary Fig. S6, S8 as:

Supplementary Figure 6. Hardware implementation of the 1PnR architecture. The UV LED array board and UV sensor array board are positioned and fixed by one-to-one for light emitting and sensing per pixel. Opaque grids are applied to avoid cross-talk between light signals of different pixels. The 5×6 optical sensors are then connected

to 30 channels of transmission circuit, which converts light information to gate voltages of memristor array for further image storage. The memristor modulation board mainly consists of FPGA core controller, array read-write circuits (ADC&DAC), WL connecting ports and switch matrix. Each WL port offers analogue connection to 30 channels of the 8k memristor array's WLS through switch matrix. And switch matrix can switch the connection of the memristor array' WLS between the WL ports for image exposure or DAC chips for data-in-situ computing.

Supplementary Figure 8. The Schematic diagram of light collection mechanism for rolling exposure experiment. Each column of the target image is reshaped to 5×6 array for exposure experiment. The received signals are reshaped back to one column through the 30-channel conversion circuits and further written to the corresponding column of memristor array in parallel using the proposed One-Pulse Method. Then, next column of image is exposed and stored to the corresponding next column, till the whole image is sensed and stored.

Corresponding change in manuscript: Yes

Location of Change:

Section 2.2: One-Pixel-Multiple-Memristor Structure and Rolling exposure strategy for fast image acquisition

Page 11:

Paragraph 3 of Section 2.2

Supplementary Figure 6

Supplementary Figure 8

Comment 2

Reviewer wrote:

Figure 3e and 3f present restored images (Weizmann silhouettes and a human portrait) that show far greater resolution and image alignment than the 5×6 sensor array could possibly provide. This implies either strong preprocessing or assumptions not disclosed in the paper.

Our response:

Thank you for your comments. As mentioned in our reply to Comment 1, we fabricated a 5×6 array of optical sensors and a corresponding UV LED array for image acquisition experiment. In our experimental setup, one column of the image pixels (up to 30 pixels) is reshaped to the array format and applied to the optical sensors via the UV LED

array. It is then reshaped back to a single column through the 30-channel conversion circuit for storage in the memristor array. Based on this, Figure 3e and 3f are preprocessed in the experiments, as described below:

Figure R3. The pre-process of the Weizmann Dataset used in experiment. The dataset has 10 actions, containing bend, jack, jump, pjump (jump in place), run, side, skip, walk, wave1 (in one hand) and wave2 (in two hands). Each action is performed by 9 people. The dataset is already binarized and aligned from video clips. In our manuscript, the dataset is cropped to 24×10 for image and network experiment. To make sure that every sample (containing 4 frames) includes a whole period of periodic actions (such as walk, run and wave), the 4 frames are averagedly picked from 12 consecutive frames in the video sequences, which are then assembled to one sample (24×40). In image acquisition experiment, 3 samples are randomly chosen from the dataset.

Figure R4. Experimental result using rolling exposure strategy for 3 samples of Weizmann dataset. Left is the target image while right is the restored image read from the memristor array.

Figure 3e shows the image acquisition experiment using the test samples in Weizmann dataset, 3 sample images are picked from the dataset for the experiment. The preprocessing procedure for the Weizmann dataset samples is illustrated in Supplementary Figure 12, also shown in Figure R3. Each sample is processed into a 24×40 -pixel image, formed by concatenating 4 consecutive 24×10 -pixel frames of human motion. During the image acquisition experiment of each sample image, 24 sensing channels are used according to the column pixels. Following the rolling exposure experiment described in Figure R1, one column (24 pixels) of the image is exposed at a time, and a total of 40 rolling exposure steps are required for the whole sample image. In fact, **Figure 3e vertically concatenates the 3 sample images (24×40) and the corresponding restored images from experimental results, whereas in the actual experiment, each sample is processed individually.** To avoid misunderstanding, we have replotted Figure 3e in the form of separated sample images, as shown in Figure R4

Figure R5. The 5 blocks separation of Rolling Exposure Experiment for the portrait image. The image is divided into 5 blocks with 30 rows of pixels to match the 30-channel circuit prepared in the experiment, The rolling exposure of 30×64 is completed within each block first, and then the next block of image is exposed in sequence. Each block of image data is stored in the corresponding block region of memristor array through switchable ports in system board.

To further evaluate the Rolling Exposure Strategy with large image size, a 128×64 portrait image is then chosen for experiment, which is shown in Figure 3f. The image size aligns with the scale of the 8K memristor array used in our study. Since the image experiment hardware can only perceive 30 pixels at a time, the full portrait image is divided into 5 blocks of 30×64 pixels (with the last block having only 8×64 pixels). Also, the memristor array is similarly divided into 5 corresponding block regions for image storage. The transmission circuits can be connected to every region’s gate lines (WLs) through switchable ports on memristor modulation board. In the experiment, The rolling exposure of 30×64 is completed within each block first, 64 exposure steps is required in each block and the image information is stores to the corresponding section of the memristor array. Then the next image block is exposed in sequence, until the entire image is processed, as shown in Figure R5.

To clarify the experiment details of the image acquisition experiment in Figure 3e and Figure 3f, **we have modified the manuscript (Section 2.2, Paragraph 4, Highlighted in yellow)** as: “RES experimental verification is firstly conducted using 3 sample images picked in the Weizmann dataset, containing actions of run, jump and walk. Each sample is processed into a 24×40-pixel image, formed by concatenating 4 consecutive 24×10-pixel frames of human motion, as illustrated in **Supplementary Fig. S12**. During the image acquisition experiment of each sample image, 24 sensing channels are used according to the column pixels, and a total of 40 rolling exposure steps are required for the whole sample image. After RES experiment, the target image and restored image from the memristor array is plotted in Fig. 3e. It can be seen from the results that the action features in restored images are still clear for classification with negligible noise.”

Also, we revise the **Supplementary Figure 9** to show the 5 blocks separation clearer, as:

Supplementary Figure 9. The schematic of the experiments for portrait image. The image is divided into 5 blocks with 30 rows of pixels to match the 30-channel circuit prepared in the experiment. The rolling exposure of 30x64 is completed within each block first, and then the next block of image is exposed in sequence. Each block of image data is stored in the corresponding block region of memristor array through switchable ports in memristor modulation board.

Corresponding change in manuscript: Yes

Location of Change:

Section 2.2: One-Pixel-Multiple-Memristor Structure and Rolling exposure strategy for fast image acquisition

Page 11:

Paragraph 4 of Section 2.2

Supplementary Figure 9

Comment 3:

Reviewer wrote:

The mechanism by which the sensed data from the ITO/ZnS/TiN pixel sensors is relayed to the memristor array via the transition circuit (shown in Fig. 3a) lacks quantitative or temporal detail. What is the timing resolution, and how is alignment ensured during the rolling exposure process?

Our response:

Thank you for your comments. The ITO/ZnS/TiN optical sensor exhibits a decrease in resistance under UV light, with greater resistance reduction under stronger illumination. Based on this characteristic, the transmission circuit we designed consists of a resistor bridge and a differential amplifier, as shown in Figure R6. The resistor bridge converts the resistance variation of the ITO/ZnS/TiN optical sensor into a voltage difference, which is then converted into the gate voltage by the differential amplifier. The potentiometer R_2 and R_G are used to adjust the reference voltage V_{ref} and the differential gain, respectively, enabling channel calibration and ensuring the circuit's output voltage range matches the transistor gate voltage requirements of the memristor array.

Figure R6. Scheme of the transmission circuit. The transmission circuit consists of a resistor bridge and a differential amplifier. The resistor bridge converts the resistance variation of the optical sensor into a voltage difference, which is then converted into the gate voltage for the memristor array by the differential amplifier.

Regarding quantitative data of the circuit, we conducted experiments on the circuit's conversion characteristics by testing the relationship between light intensity and the circuit's output voltage. The results are presented in Supplementary Figure 5, also shown in Figure R7a. We selected 9 test points within the light intensity range and recorded the corresponding output voltages. Fitting analysis shows that the relationship between light intensity and output voltage follows a power function: $I = a \cdot D^b + c$. Supplementary Figure 2d already indicates that the response of the optical sensor to light intensity also exhibits a power-function relationship. Since the differential amplifier circuit performs a linear transformation, the fitting results of the entire output circuit correspond to the characteristics of the optical sensor, further confirming the consistency and rationality of our circuit test results.

Figure R7. The transition characteristic of the transmission circuit. **a** the relationship of the transmission circuit from light density to output voltage. The black spot is the measured data, and the purple curve is the fitted result with power function relationship: $I = a \cdot D^b + c$. The circuit process is linear, so the fitted relationship is consistent with the sensor's characteristic. **b** the experiment setup of transmission speed test. A signal generator is connected to the input ports of the differential amplifier to provide the reference voltage V_{ref} and the voltage division V_{OR} on the optical sensor, while the oscilloscope simultaneously monitors the V_{OR} and the output voltage V_{out} of the circuit. **c** The experiment result of transmission speed test, indicating ~920 ns transmission time.

Concerning the temporal parameters, since the LEDs are controlled by powering MOS transistors, we find from experiments that the settling time - from applying the control signal to the LED reaching maximum intensity - is on the millisecond scale. Therefore, it is difficult to accurately measure the response time of the optical sensor and conversion circuit using the prepared LED board. Instead, we evaluated the circuit's temporal performance through alternative experiments. First, the photoelectric effect in the ZnS device is nearly instantaneous, so it can be considered negligible^[1]. Second, the speed of the differential amplifier circuit itself is tested using a signal generator and an oscilloscope. A simplified model of the conversion circuit is shown in Figure R7b. The signal generator is connected to the input ports of the differential amplifier to provide the reference voltage V_{ref} and the voltage division V_{OR} on the optical sensor, while the oscilloscope simultaneously monitors the V_{OR} and the output voltage V_{out} of the circuit. The captured waveform of the speed test is shown in Figure R7c. The yellow curve represents the voltage division V_{OR} of the optical sensor, the falling portion is focused because V_{OR} drops under UV illumination. The green curve represents the output voltage of the differential circuit, which rises due to the decrease in the negative input. The results indicate that the time from the input voltage drop to the stabilization of the output voltage rise is about 920 ns, which reflects the speed of our conversion circuit. It should be noted that this delay is measured at the board level. With future chip-level integration, where parasitic capacitance and other non-ideal effects can be greatly reduced, we believe the circuit speed can be further improved.

Figure R8. The timing sequence 1PnR hardware system in image acquisition experiment. Optical signals are first generated by LED array, then 1PnR system receives the signal and stores them to memristor array using the proposed rolling exposure strategy.

Regarding timing alignment during the rolling exposure process, according to our experimental design and system setup, the sequence for each rolling exposure operation is as follows: The PC host computer converts the grayscale values of the 30 pixels to be exposed into control voltages for the LED-driving MOS transistors. These values are transmitted via serial port to the MCU, which controls the DAC chips to output the corresponding voltage signals. Since serial communication, multi-channel DAC control, and UV LED stabilization all require responding time, a short time delay is applied for stabilization. Subsequently, the PC host computer sends the write signal to the memristor chip board system, the system then applies the One-Pulse-Method, enabling parallel writing of the 30-channel optical sensing information into the array. The system timing sequence is summarized in Figure R8.

References:

[1] T. Sun. (2003). Two-photon absorption autocorrelation of visible to ultraviolet femtosecond laser pulses using ZnS-based photodetectors. IEEE Photonics Technology Letters, 14(1), 86-88.

To clarify timing alignment of the rolling exposure experiment, **we have modified the manuscript (Section 2.2, Paragraph 1, Highlighted in yellow)** as: “**The circuit parameter is adjusted to fit the gate voltage range of the 1T1R array, and the transmission relationship of the circuit from light density to gate voltage, as well as the transmission speed is plotted in Supplementary Fig. S5.**”, and the manuscript (Section 2.2, Paragraph 3, Highlighted in yellow) as: “**The image is converted into UV light signals based on pixel values and directly irradiated to the optical sensors, which simulates the optical exposure process. And the detailed process of the exposure experiment based on the hardware system is illustrated in Supplementary Fig. S7.**”, as well as the corresponding Supplementary Fig. S5, S7 as:

Supplementary Figure 5. a the relationship of the transmission circuit from light density to output voltage. The black spot is the measured data, and the purple curve is the fitted result with power function relationship: $I = a \cdot Db + c$. The circuit process is purely linear, so the fitted relationship is consistent with the sensor’s characteristic shown in Supplementary Fig. S2d. Furthermore, the output voltage of the circuit is adjusted to fit gate voltage of the 1T1R cell. **b The responding speed of the transmission circuit.** A signal generator is connected to the input ports of the differential amplifier to provide the signal, while an oscilloscope simultaneously monitors the input and output signal of the circuit. The yellow curve represents the voltage division VOR of the optical sensor, the falling portion is focused because VOR drops under UV illumination. The green curve represents the output voltage of the differential circuit, which rises due to the decrease in the negative input. The results indicate that the time from the input voltage drop to the stabilization of the output voltage rise is about 920 ns, which reflects the speed of our conversion circuit. It should be noted that this delay is measured at the board level. With future chip-level integration, where parasitic capacitance and other non-ideal effects can be greatly reduced, we believe the circuit speed can be further improved.

Supplementary Figure 7. the system timing sequence in image acquisition experiment. The PC host computer converts the grayscale values of the 30 pixels to be exposed into control voltages for the LED-driving MOS transistors. These values are transmitted via serial port to the MCU, which controls the DAC chips to output the corresponding voltage signals. Subsequently, the PC host computer sends the write signal to the memristor chip board system, the system then applies the One-Pulse-Method, enabling parallel writing of the 30-channel optical sensing information into the array.

Corresponding change in manuscript: Yes

Location of Change:

Section 2.2: One-Pixel-Multiple-Memristor Structure and Rolling exposure strategy for fast image acquisition

Page 9:

Paragraph 1 of Section 2.2

Page 11:

Paragraph 3 of Section 2.2

Supplementary Figure 5

Supplementary Figure 7

Comment 4:

Reviewer wrote:

For Figure 3f (portrait image), Supplementary Figure 5 explains a division into 5 blocks of 30×64 segments, but again, there is no mention of the optical projection method, scene conditions, or how misalignment or motion was controlled. This raises concerns that the "image" may be more synthetic or pattern-based than optically formed.

Our response:

Thank you for your comments. As mentioned in our previous responses, this experiment does not employ a lens system for physical object exposure. Instead, it utilizes a one-to-one optical mapping between the UV LED array and the optical sensor array, directly converting digital pixel values into corresponding light signals projected onto the sensors. This approach essentially simulates ideal exposure conditions by controlling LED brightness, thereby avoiding issues such as scene variation, motion blur, or alignment errors which may arise in actual optical projection systems. This setup allows us to focus more on validating the 1PnR architecture and the data-in-situ computing network.

Furthermore, based on the constructed hardware system, we have provided a detailed explanation of the principle for dividing the portrait image in Figure 3f into 5 blocks for rolling exposure experiments. A clearer schematic diagram has also been included to illustrate this process, as shown in Figure R5 above.

Figure R9. Prospects of the 1PnR system for future image exposure. Multiple memristor arrays and Large scale sensors are employed, each column of sensors can be connected to a different memristor array. After exposure, write pulses can be applied simultaneously to different arrays, enabling the entire image to be written into separate memristor arrays at the same time, offering very fast image exposure.

Regarding the potential optical imaging applications, we'd like to discuss the prospects of our proposed 1PnR architecture. In fact, the 1PnR sensing-storage architecture exhibits remarkable flexibility and scalability. For future optical system applications, when the sensor array scale becomes sufficiently large, it can function as an area sensor for global exposure. Following the operational principle of line-scan CCD cameras, the entire image can be first exposed globally, and then its pixels can be transferred column by column to the memristor array for storage. If multiple memristor arrays are employed, each column of sensors can be connected to a different memristor array. After exposure, write pulses can be applied simultaneously to different arrays, enabling the entire image to be written into separate memristor arrays at the same time, offering very fast image exposure, as illustrated in Figure R9. Moreover, if the scale of the memristor array is sufficiently large, all sensor units in the area array can be connected to the same array, allowing an entire image to be written into a single column of memristors.

To clarify this, we have added a discussion on prospects of 1PnR architecture for future optical system applications, and modified the manuscript (Section 2.2, Paragraph 6, Highlighted in yellow) as: "In fact, the proposed 1PnR architecture exhibits remarkable flexibility and scalability for image acquisition. For future optical system applications, when the sensor array scale becomes sufficiently large, it can function as an area sensor for global exposure. Following the operational principle of line-scan CCD cameras, the entire image can be first exposed globally, and then its pixels can be transferred column by column to the memristor array for storage. If multiple memristor arrays are employed, each column of sensors can be connected to a different memristor array. After exposure, write pulses can be applied simultaneously to different arrays, enabling the entire image to be written into separate memristor arrays at the same time, offering very fast image exposure, as illustrated in **Supplementary Fig. S9**. Moreover, if the scale of the memristor array is sufficiently large, all sensor units in the area array can be connected to the same array, allowing an entire image to be written into a single column of memristors.", and added **Supplementary Fig. S10** as:

Supplementary Figure 10. Prospects of the 1PnR system for future image exposure. Multiple memristor arrays and large-scale sensors are employed, each column of sensors can be connected to a different memristor array. After exposure, write pulses can be applied simultaneously to different arrays, enabling the entire image to be written into separate memristor arrays at the same time, offering very fast image exposure.

Corresponding change in manuscript: Yes

Location of Change:

Section 2.2

Page 12:

Paragraph 6 of Section 2.2

Supplementary Figure 10

Comment 5:

Reviewer wrote:

The claim that a 24×40 image from the Weizmann dataset can be reconstructed column by column assumes each column corresponds directly to an optical sensor signal. However, the paper doesn't clarify whether a lens forms a spatial image or if each sensor merely encodes a temporal sequence. Without this, the legitimacy of the image modality is uncertain.

Our response:

Thank you for your comments. The experimental setup and principle of our image exposure experiment have been explained in our previous responses. Briefly, instead of using an optical lens system to capture actual objects, we employ a one-to-one correspondence between the UV LED array and the optical sensor array to directly convert pixel values of the sample images into corresponding optical signals and illuminate to sensors.

Figure R10. The pre-process of the Weizmann Dataset used in experiment. The dataset has 10 actions, containing bend, jack, jump, pjump (jump in place), run, side, skip, walk, wave1 (in one hand) and wave2 (in two hands). Each action is performed by 9 people. The dataset is already binarized and aligned from video clips. In our manuscript, the dataset is cropped to 24×10 for image and network experiment. To make sure that every sample (containing 4 frames) includes a whole period of periodic actions (such as walk, run and wave), the 4 frames are averagely picked from 12 consecutive frames in the video sequences, which are then assembled to one sample (24×40).

As for the sample image of human action, each sample is formed by concatenating 4 consecutive images of human motion in Weizmann dataset. This essentially simulates capturing 4 snapshots at different time points within a continuous action sequence. The sample processing procedure for the Weizmann dataset is illustrated in Supplementary Figure 12, also shown in Figure R10. To make sure that every sample (containing 4 frames) includes a whole period of periodic actions (such as walk, run and wave), the 4 frames are averagely picked from 12 consecutive frames in the video sequences.

Corresponding change in manuscript: No

Comment 6:

Reviewer wrote:

The object-imaging chain — from real-world object, through optical transduction, to data conversion and final classification — is not sufficiently detailed to confirm the system's operation as a real imaging device rather than a neuromorphic signal processor using artificially prestructured inputs.

Our response:

Thank you for your comments. As previously addressed in our responses, we have introduced the hardware implementation of the image exposure experiment. Here, we present the overall system diagram of the 1PnR system, covering the entire process from image sensing to computation, as shown in Figure R11. The system consists of a PC host computer, an LED control board, an LED array board, an optical sensor array, an analog domain conversion circuit, and a memristor modulation board. The functions of each module and the signal transmission process within the system are furthermore illustrated in Figure R12.

As can be observed, the entire system can be divided into two parts: the optical signal emitting part and the

image sensing-storage-computation part based on the 1PnR architecture. In the optical signal emitting part, the PC host computer converts the grayscale values of the pixel columns to be exposed into control voltages for the LED-driving MOS transistors. These signals are transmitted via serial communication to the MCU, which controls the DAC chips to output corresponding voltage signals, thereby driving the LED array to emit the corresponding optical signals. In the 1PnR system part, the optical sensor array receives the optical signals, which are then converted in parallel into gate voltages for the memristor array through the multi-channel conversion circuit. Subsequently, the memristor modulation board receives commands from the PC host computer and writes the image data into the corresponding regions of the array. Additionally, the memristor modulation board mainly consists of FPGA core controller, array read-write circuits (ADC&DAC), WL connecting ports and switch matrix. Each WL port offers analogue connection to 30 channels of the 8k memristor array's WLs through switch matrix. And switch matrix can switch the connection of the memristor array's WLs between the WL ports for image exposure or DAC chips for data-in-situ computing. This achieves full chain validation of image from sensing and storage to computation.

It should be noted that the 1PnR verification system we constructed integrates the sensors, transmission circuits, and the memristor array which can be interconnected via analog circuits. While it is currently implemented as a board-level validation system, it is designed with the potential for future chip-level integration. Therefore, it should be regarded as an integrated system for image sensing, storage, and computation, rather than merely a neuromorphic computing processor.

Figure R11. Hardware implementation of the 1PnR architecture. The UV LED array board and UV sensor array board are positioned and fixed by one-to-one for light emitting and sensing per pixel. Opaque grids are applied to avoid cross-talk between light signals of different pixels. The 5×6 optical sensors are then connected to 30 channels of transmission circuit, which converts light information to gate voltages of memristor array for further image storage. The memristor modulation board mainly consists of FPGA core controller, array read-write circuits (ADC&DAC), WL connecting ports and switch matrix. Each WL port offers analogue connection to 30 channels of the 8k memristor array's WLs through switch matrix. And switch matrix can switch the connection of the memristor array's WLs

between the WL ports for image exposure or DAC chips for data-in-situ computing.

Figure R12. The functions of each module and the signal transmission process of the 1PnR hardware system. The entire system can be divided into two parts: the optical signal emitting part and the image sensing-storage-computation part based on the 1PnR architecture.

To clarify this, we have added an overall hardware picture in the Figure 1 and provided a detailed introduction of the 1PnR hardware implementation, and **modified the manuscript (Section 2.2, Paragraph 3, Highlighted in yellow)** as: “Utilizing the fabricated pixel sensors and transmission circuit, a 1PnR hardware verification system is implemented, as shown in **Supplementary Fig. S6**. UV LED light sources are employed, corresponding one-to-one with the fabricated pixel sensors. The image is converted into UV light signals based on pixel values and directly irradiated to the optical sensors, which simulates the optical exposure process,” and modulated Figure 1, Supplementary Fig. S6 as:

Fig. 1. Schematic of the human visual system and the proposed artificial visual system based on 1PnR structure. The human vision system contains eyes, optic neurons and brain, the visual working memory in human brain stores temporary sequential visual information and performs pre-process. The proposed artificial visual system based on 1PnR architecture is composed of ITO/ZnS/TiN pixel sensors for image sensing, analogue circuit for data transmission, and an 8k memristor array based on TiN/HfO_x/TaO_x/TiN stack. The array is divided into 2 parts, one part serves as working memory, which stores image data and performs data-in-situ for pre-process. The other part serves as neuromorphic computing core, which stores network weights and carries network computing for classification.

Supplementary Figure 6. Hardware implementation of the 1PnR architecture. The UV LED array board and UV sensor array board are positioned and fixed by one-to-one for light emitting and sensing per pixel. Opaque grids are applied to avoid cross-talk between light signals of different pixels. The 5×6 optical sensors are then connected to 30 channels of transmission circuit, which converts light information to gate voltages of memristor array for further image storage. The memristor modulation board mainly consists of FPGA core controller, array read-write circuits (ADC&DAC), WL connecting ports and switch matrix. Each WL port offers analogue connection to 30 channels of the 8k memristor array's WLs through switch matrix. And switch matrix can switch the connection of the memristor array' WLs between the WL ports for image exposure or DAC chips for data-in-situ computing.

Corresponding change in manuscript: Yes

Location of Change:

Figure 1

Section 2.2: One-Pixel-Multiple-Memristor Structure and Rolling exposure strategy for fast image acquisition

Page 11:

Paragraph 3 of Section 2.2

Supplementary Figure 6

Reviewer #1 (Remarks on code availability):

Since I am a hardware engineer, I am not in the best position to evaluate the provided source code. However, I recommend referring to my review regarding the measurements and data interpretation.

Our response:

We sincerely thank Reviewer #1 for the careful comments and valuable feedback on the hardware system implementation and image acquisition experiment, which have helped us significantly improve our manuscript. We have provided detailed responses to all related questions in our corresponding replies.

The source code provided serves as a demonstration of our proposed memristor-based data-in-situ computing network. Unlike traditional memristor neural networks that require image data input, our data-in-situ computing network directly delivers network weights to the image stored memristor arrays, enabling image processing at the data storage location. This approach eliminates image data readout and transfer, making it particularly suitable for sensing-storage-computing integrated architectures. In this demonstration, we utilized the standard MNIST dataset and achieved a recognition rate of 95.9%, validating the classification capability of the data-in-situ computing network.

We have verified that the code executes correctly and can reproduce the reported results.

Reviewer #2 (Remarks to the Author):

This paper introduces a one-pixel-multiple-memristor(1PnR) architecture to enhance image processing efficiency. By connecting each pixel to multiple memristors, the authors enable parallel storage and in-situ computing within a memristor array, significantly reducing data transmission overhead. The proposed system incorporated a rolling exposure strategy for high-speed image acquisition and reported 95.7% accuracy on the Weizmann human action dataset. Additionally, the system showed improvements in latency and energy consumption compared to conventional CMOS-based systems.

However, despite the practical demonstration, this manuscript suffers from insufficient explanation.

Details are elaborated below:

1) The claimed benefits of the 1PnR architecture over previously reported sensor-memory-compute integrated systems are not clearly demonstrated. Quantitative or qualitative comparisons are necessary to justify the advantage of this architecture beyond the reported speed and power gains.

2) The manuscript lacks a detailed discussion on how the proposed architecture could be scaled or implemented as fully integrated system.

With respect to the concerns outlined above, the following issues should be addressed:

Our Response to Reviewer #2:

We sincerely thank Reviewer #2 for the careful review, as well as their positive assessment of our work. We have carefully considered all the comments and provide detailed responses below.

Comment #1:

Reviewer wrote:

In 'Introduction', The manuscript provides a compelling vision for neuromorphic visual processing; however, the novelty of the proposed 1PnR architecture in comparison to existing sensor-memory-compute integrated systems—such as 2D/3D-stacked neuromorphic imagers or hybrid CMOS-memristor solutions—is not sufficiently highlighted. Including a more explicit architectural or performance comparison with recent state-of-the-art systems

would help better situate this work within the current research landscape and substantiate its unique contributions.

Our response:

Thank you for your comments. In the Introduction, we briefly mentioned the current sensing-storage-computation integrated architectures and their existing challenges. To more clearly highlight the innovations of our work, we list 3 representative works about integrated sensing-storage-computation systems (1PT1R, MPT1R, 1T1OR architecture) and conduct a detailed comparison with our proposed 1PnR architecture, as shown in Table R1.

Table R1. Comparison between the 1PnR architecture and existing sensor-memory integrated architecture

	Integrating Architecture	Verified Scale	Supported Network	Performance	Power Consumption	Exposure Time	Data Erase ^[a]	Data Transfer ^[b]
Adv. Mater. 2022 ^[1]	1PT1R	16×3-PT	OANN	99.3% (“P” “K” “U”)	/	/	Yes	No
Nat. Electron. 2024 ^[2]	MPT1R	20×20-PT 36×32-1T1R	OCNN ORNN OSNN	91% (MNIST)	/	/	Yes	No
Nat. Nanotechnol 2024 ^[3]	1T1OR	128×8-1T1OR	ORNN	91.2% (NTU-RGB dataset)	24.8 nJ ^[c]	/	Yes	Yes
This Work	1PnR	5×6-Pixel 64×128-1T1R	Data-in-situ Computing	95.7% (Weizmann dataset)	581.73 pJ ^[d]	12 μs ^[e]	No	No

[a] Whether the storage device (memristor or optoelectronic memristor) needs to be erased before receiving next frame data in sequential vision.

[b] Whether the full image data needs to be transferred during computing process.

[c] The estimated energy consumption of the proposed OEM-based RC for one sample classification in human action recognition.

[d] The estimated energy consumption of the data-in-situ architecture for one sample in Weizmann dataset, which has about 160 times energy efficiency than typical CMOS systems.

[e] The estimated time consumption for processing 120×120 image using the 1PnR architecture, which exhibits ~2000 times reduction than CMOS sensors.

It should be noted that current comparative works usually focus on validating new integrating structures, and some of the specific timing and power consumption data are not provided. Moreover, due to the complexity of chip structures and operational principles, different algorithms and strategies significantly impact timing and power consumption, making it difficult to provide accurate parameter estimations. Therefore, we limit our comparison here to a qualitative analysis based on integrating architecture and the working principles. At the same time, we provide timing and power consumption parameters of our system, offering quantitative references for evaluating the performance of the 1PnR system and for future comparisons.

Based on the comparative results with other works, we identify the following advantages of our architecture:

First, it supports continuous sensing and storage of sequential images. With one sensor corresponding to multiple storage units, sensing information can be rapidly and sequentially written into different storage units without erasing previously stored image data.

Second, it offers great scalability for large-scale integration. The architecture leverages relatively mature technologies for integrating optical sensors with memristor arrays. Since the sensors are directly connected to the gates of the memristor array, no modification of the internal structure of the memristor array is required, enabling straightforward integration. The image storage capacity can be easily expanded by simply increasing the scale of the memristor array.

Third, it provides high flexibility for system expansion. The independent nature of the sensor units and the memristor array allows for versatile configurations. For example, the sensor array can be integrated with multiple memristor arrays, where each column of sensors is connected to a different memristor array. After global exposure, writing pulses can be applied simultaneously to all arrays, enabling the entire image to be written at once and

achieving ultra-high-speed image acquisition.

References:

[1] Dang, B. (2023). One-Phototransistor-One-Memristor Array with High-Linearity Light-Tunable Weight for Optic Neuromorphic Computing. *Advanced Materials*, 35(37), 2204844.

[2] Dang, B. (2024). Reconfigurable in-sensor processing based on a multi-phototransistor-one-memristor array. *Nature Electronics*, 7, 991-1003.

[3] Huang, H. (2024). Fully integrated multi-mode optoelectronic memristor array for diversified in-sensor computing. *Nature Nanotechnology*, 20, 93-103.

To clarify this, we have modified the manuscript (Section 1, Paragraph 2, Highlighted in yellow) as: “In addition, the sensor and memristor device is usually integrated as one-sensor-one-memristor²⁸⁻³⁰, or multi-sensor-one-memristor structure^{31,32}. In sequential image processing conditions, read-out and erasing operation is required for previously stored image data before sensing new pixel data, which limits the image sensing speed and energy efficiency”, and modified the manuscript (Section 2.3, Paragraph 8, Highlighted in yellow) as: “Finally, the comparison of the 1PnR system and other representative works about integrated sensing-storage-computation system is performed and summarized in Supplementary Note 4.”, also added Supplementary Note 4, Supplementary Table 4 as:

Supplementary Note 4. Comparison between the 1PnR architecture and existing sensor-memory integrated architecture

To highlight the innovations of the 1PnR architecture, 3 representative works about integrated sensing-storage-computation systems (1PT1R, MPT1R, 1T1OR architecture) are listed and compared in Supplementary Table 4.

Supplementary Table 4. Comparison between the 1PnR architecture and existing sensor-memory integrated architecture

	Integrating Architecture	Verified Scale	Supported Network	Performance	Power Consumption	Exposure Time	Data Erase ^[a]	Data Transfer ^[b]
Adv. Mater. 2022 ^[1]	1PT1R	16×3-PT	OANN	99.3% (“P” “K” “U”)	/	/	Yes	No
Nat. Electron. 2024 ^[2]	MPT1R	20×20-PT 36×32-1T1R	OCNN ORNN OSNN	91% (MNIST)	/	/	Yes	No
Nat. Nanotechnol 2024 ^[3]	1T1OR	128×8-1T1OR	ORNN	91.2% (NTU-RGB dataset)	24.8 nJ ^[c]	/	Yes	Yes
This Work	1PnR	5×6-Pixel 64×128-1T1R	Data-in-situ Computing	95.7% (Weizmann dataset)	581.73 pJ ^[d]	12 μs ^[e]	No	No

[a] Whether the storage device (memristor or optoelectronic memristor) needs to be erased before receiving next frame data in sequential vision.

[b] Whether the full image data needs to be transferred during computing process.

[c] The estimated energy consumption of the proposed OEM-based RC for one sample classification in human action recognition.

[d] The estimated energy consumption of the data-in-situ architecture for one sample in Weizmann dataset, which has about 160 times energy efficiency than typical CMOS systems.

[e] The estimated time consumption for processing 120×120 image using the 1PnR architecture, which exhibits ~2000 times reduction than CMOS sensors.

Based on the comparative results with other works, the 1PnR architecture can be identified the following advantages:

First, it supports continuous sensing and storage of sequential images. With one sensor corresponding to multiple storage units, sensing information can be rapidly and sequentially written into different storage units without erasing previously stored image data.

Second, it offers great scalability for large-scale integration. The architecture leverages relatively mature technologies for integrating optical sensors with memristor arrays. Since the sensors are directly connected to the

gates of the memristor array, no modification of the internal structure of the memristor array is required, enabling straightforward integration. The image storage capacity can be easily expanded by simply increasing the scale of the memristor array.

Third, it provides high flexibility for system expansion. The independent nature of the sensor units and the memristor array allows for versatile configurations. For example, the sensor array can be integrated with multiple memristor arrays, where each column of sensors is connected to a different memristor array. After global exposure, writing pulses can be applied simultaneously to all arrays, enabling the entire image to be written at once and achieving ultra-high-speed image acquisition.

Corresponding change in manuscript: Yes

Location of Change:

Section 1: Introduction

Page 4:

Paragraph 2 of Section 1

Section 2.3: Data-in-situ computing network for neuromorphic sequential vision

Page 19:

Paragraph 8 of Section 2.3

Supplementary Note 4

Supplementary Table 4

Comment #2:

Reviewer wrote:

In 'Date-in-situ computing network for neuromorphic sequential vision' section, While the use of the Weizmann human action flow dataset demonstrates the classification capability of the proposed system, it would be beneficial to evaluate the model on more complex and diverse datasets. In particular, testing under real-world variations such as lighting changes, occlusions, and sensor noise could better establish the robustness and generalizability of the 1PnR architecture.

Our response:

Thank you for your comments. In addition to the human motion dataset, we also conducted experimental validation using the standard MNIST dataset to verify the general classification capability of our system. The experimental process and results are presented in Supplementary Figure 13 and Supplementary Note 3. The results show that our system achieves recognition rates of 95.9% (simulation) and 92.2% (experimental measurement) on the MNIST dataset, further demonstrating the classification ability of the proposed architecture.

As suggested by the reviewer, we further performed simulations to investigate the potential noise in real-world variations, validating the robustness of data-in-situ network. The results are shown in Figure R13. Figure R13a illustrates the network recognition accuracy for MNIST dataset with no noise, a final recognition rate of 95.9% is achieved, which is considered as reference. Figure R13b shows the simulation of the influence of image brightness on recognition accuracy. To simulate different brightness variations, the grayscale values are mapped to the range $[0, 1]$. For over-dark variation, the grayscale values were compressed to $[0, k]$, while for over-bright variation, they were compressed to $[k, 1]$. The results indicate that over-dark variation has almost no effect on recognition accuracy, Indeed the sensor sensitivity should be sufficiently high. In contrast, over-bright variation has a significant impact on accuracy, due to the loss of dark-level information. Figure R13c presents the simulation results of image sensor noise, which is applied to the image data to simulate the noise of the image sensor. The noise is applied in randomly

generated normal distribution, with mean value of 1. Each standard deviation is simulated for 100 cycles. Figure R13d shows the simulation of image sensor yield rate, random bad pixels are applied according to the yield rate. It can be observed that the network maintains relatively good recognition performance at sensor noise of $\sigma = 0.2$ or a 90% sensor yield rate, revealing the good robustness of the data-in-situ network.

Figure R13. The sensor noise experiment based on MNIST dataset. **a** the network recognition accuracy for MNIST dataset with no noise. **b-d** the simulation of the influence of image brightness, sensor noise and sensor yield on recognition accuracy.

Figure R14. Simulation of data-in-situ computing network for Fashion MNIST. **a** the schematic diagram of the data-in-situ computing network for Fashion MNIST. Fashion MNIST has more complex features, 4 Voltage Vectors are applied to extract more image features, then attaching $112 \times 80 \times 10$ network for further classification. **b** the network recognition accuracy for Fashion MNIST dataset with no noise. **c-e** the simulation of the influence of image

brightness, sensor noise and sensor yield on recognition accuracy.

Furthermore, following the reviewer's suggestion, we also performed simulation validation using the more complex Fashion MNIST dataset. The schematic diagram of the data-in-situ computing network for Fashion MNIST dataset is shown in Figure R14a. Given the greater complexity of Fashion MNIST images compared to MNIST, we applied 4 Voltage Vectors to the image for data-in-situ computing to extract more image features. The data-in-situ computing results of 4 voltage vectors are concatenated as 112 feature values, which are then fed into a $112 \times 80 \times 10$ network for further classification. The network's recognition accuracy reaches 87.17%, as shown in Figure R14b. Additionally, we conducted the same noise experiments as in Figure R13 and obtained similar robust test results, as shown in Figure R14c-e.

The results from these experiments can serve as valuable supplementary evidence, further validating the network's recognition capability and robustness.

To clarify this, we have modified the manuscript (Section 2.3, Paragraph 8, Highlighted in yellow) as: To evaluate the general classification ability of data-in-situ computing network, the benchmark is then performed on the commonly used MNIST and Fashion MNIST dataset with noise analysis. The experimental results on MNIST dataset are presented in Supplementary Fig. S13 and Supplementary Note 3, which demonstrates a 95.9% (simulation) and 92.4% (experimental) classification accuracy. The simulation results on Fashion MNIST dataset are shown in Supplementary Fig. S14. Given the greater complexity of Fashion MNIST images compared to MNIST, 4 voltage vectors are applied to the image for data-in-situ computing to extract more image features, which achieves an 87.17% recognition accuracy. These results further demonstrate the great performance of the data-in-situ computing network for future neuromorphic visual systems.

Also, the Supplementary Fig. S13 and Supplementary Fig. S14 are modified as:

Supplementary Figure 13. Experimental demonstration of the MNIST recognition based on 1PnR hardware system. **a** schematic of the data-in-situ network for MNIST recognition. Two voltage weight vectors (VWV) are applied to perform data-in-situ computing, and $56 \times 40 \times 10$ read-out layer is used for classification. **b** the simulated

results of the network. When 2 VWVs are both applied from the row of the memristor, the classification accuracy reaches best to 95.9%, if 2 VWVs are both applied from column of the array, the accuracy reduces to 91.7%. The MNIST samples have more features after data-in-situ computing in rows direction and achieve better classification performance. **c** the deployment of the network in 8k memristor array. Color blocks reveal the partitions for layer mapping. **d** the Weight-transfer errors of the HD (56×40) and FC (40×10) layers to the memristor array by differential group. The transfer errors are limited within 6 μ S by an array modulation script. **e** the recognition results for MNIST dataset of the hardware system, with 92.42% accuracy. **f** the simulation of the influence of image brightness on recognition accuracy. The grayscale values are mapped to the range [0, 1]. For over-dark variation, the grayscale values were compressed to [0, k], while for over-bright variation, they were compressed to [k, 1]. Over-dark variation has almost no effect on recognition accuracy, the sensor sensitivity should be sufficiently high. In contrast, over-bright variation has a significant impact on accuracy, due to the loss of dark-level information. **g** the simulation of image sensor noise. The noise is applied in randomly generated normal distribution, with mean value of 1. Each standard deviation is simulated for 100 cycles. **h** the simulation of image sensor yield rate, random bad pixels are applied according to the yield rate. It can be observed that the network maintains relatively good recognition performance at sensor noise of $\sigma = 0.2$ or a 90% sensor yield rate, revealing the good robustness of the data-in-situ network.

Supplementary Figure 14. Simulation of data-in-situ computing network for Fashion MNIST. a the schematic diagram of the data-in-situ computing network for Fashion MNIST. Fashion MNIST has more complex features, 4 Voltage Vectors are applied to extract more image features, then attaching 112×80×10 network for further classification. **b** the network recognition accuracy for Fashion MNIST dataset with no noise. **c-e** the simulation of the influence of image brightness, sensor noise and sensor yield on recognition accuracy.

Corresponding change in manuscript: Yes

Location of Change:

Section 2.3: Data-in-situ computing network for neuromorphic sequential vision

Page 19:

Paragraph 8 of Section 2.3

Supplementary Figure 13

Supplementary Figure 14

Comment #3:

Reviewer wrote:

The manuscript presents a promising hardware implementation of the 1PnR system. However, it remains unclear whether the entire system—including the optical sensor, analog front-end, memristor array, and in-situ computation unit—has been physically fabricated and tested as an integrated platform. A detailed description of the experimental setup, along with discussions on system-level limitations such as scalability, endurance, and noise resilience of the memristor devices, would significantly strengthen the claims on practical viability.

Our response:

Thank you for your comments. We only provided a brief description of the system setup in the initial version of the manuscript, and we apologize for any misunderstanding. Here, we provide a detailed explanation of the entire experimental system.

1. Overall Hardware Implementation of the 1PnR Architecture

The overall experimental implementation of the 1PnR architecture is illustrated in Figure R15. The system consists of a PC host computer, an LED control board, an LED array board, an optical sensor array, an analog domain conversion circuit, and a memristor modulation board. The functions of each module and the signal transmission process within the system are furthermore illustrated in Figure R16. As can be observed, the entire system can be divided into two parts: the optical signal emitting part and the image sensing-storage-computation part based on the 1PnR architecture.

Figure R15. Hardware implementation of the 1PnR architecture. The UV LED array board and UV sensor array board are positioned and fixed by one-to-one for light emitting and sensing per pixel. Opaque grids are applied to avoid cross-talk between light signals of different pixels. The 5×6 optical sensors are then connected to 30 channels of transmission circuit, which converts light information to gate voltages of memristor array for further image storage.

The memristor modulation board mainly consists of FPGA core controller, array read-write circuits (ADC&DAC), WL connecting ports and switch matrix. Each WL port offers analogue connection to 30 channels of the 8k memristor array's WLs through switch matrix. And switch matrix can switch the connection of the memristor array's WLs between the WL ports for image exposure or DAC chips for data-in-situ computing.

Figure R16. The functions of each module and the signal transmission process of the 1PnR hardware system. The entire system can be divided into two parts: the optical signal emitting part and the image sensing-storage-computation part based on the 1PnR architecture.

In the optical signal emitting part, the PC host computer converts the grayscale values of the pixel columns to be exposed into control voltages for the LED-driving MOS transistors. These signals are transmitted via serial communication to the MCU, which controls the DAC chips to output corresponding voltage signals, thereby driving the LED array to emit the corresponding optical signals.

In the 1PnR system part, the optical sensor array receives the optical signals, which are then converted in parallel into gate voltages for the memristor array through the multi-channel conversion circuit. Subsequently, the memristor modulation board receives commands from the PC host computer and writes the image data into the corresponding regions of the array. Additionally, the memristor modulation board mainly consists of FPGA core controller, array read-write circuits (ADC&DAC), WL connecting ports and switch matrix. Each WL port offers analogue connection to 30 channels of the 8k memristor array's WLs through switch matrix. And switch matrix can switch the connection of the memristor array's WLs between the WL ports for image exposure or DAC chips for data-in-situ computing. This achieves full chain validation of image from sensing and storage to computation.

2. Hardware Implementation of the Image Acquisition Experiment

In this manuscript, the image acquisition hardware is designed with one-to-one optical mapping between the UV LED array and the optical sensor array, the image is converted into UV light signals and directly irradiated to the optical sensors to simulate the optical exposure process and verify the 1PnR architecture. The details are as follows:

In image acquisition hardware, a 5×6 optical sensor array is fabricated on a 4-inch silicon wafer, and a 5×6 UV LED array board is designed corresponding one-to-one with the optical sensors, enabling pixel-by-pixel light signal transmission and sensing. Additionally, 3D-printed light-opaque grids are applied to isolate different LED light sources and optical sensors, preventing cross-talk between light signals of different pixels. The LED light source board and the optical sensor array board are precisely aligned and fixed using 3D-printed interlocking parts. The

electrodes of each sensor are connected via copper wires to the PCB baseboard and then linked to the gates of the memristor array through conversion circuits. Corresponding to the number of optical sensors, the board contains 30 channels of conversion circuits, which can convert the light information simultaneously to gate voltages of the memristor array for image storage.

Figure R17. The Schematic diagram of light collection mechanism for rolling exposure strategy. Each column of the target image is reshaped to 5×6 array for exposure experiment. The received signals are reshaped back to one column through the 30-channel conversion circuits and further written to the corresponding column of memristor array in parallel using the proposed One-Pulse Method. Then, next column of image is exposed and stored to the corresponding next column, till the whole image is sensed and stored.

During the image acquisition experiment, reshaping of the image column is applied, as shown in Figure R17. In the manuscript, the proposed rolling exposure strategy senses and stores the image column by column. To achieve experimental verification with larger image size based on the prepared image acquisition hardware, each column of the image (30 pixels) is reshaped to a 5×6 array and is applied to the optical sensor array via the UV LED array. The light signals received by the optical sensors are then reshaped back to one column through the 30-channel conversion circuits and connected to the gates (WLs) of the 1T1R memristor array. The image column data is further stored in the corresponding column of the memristor array in parallel using the proposed One-Pulse Method. Then, next column of image is exposed and stored to the corresponding next column, till the whole image is sensed and stored. This experimental approach allows us to achieve verification of rolling exposure strategy with larger image size as effectively as possible.

3. Discussions On System-Level Limitations of the 1PnR Architecture

As suggested by the reviewer, we have made in-depth analysis on system-level limitations of the 1PnR architecture based on the hardware implementation in our experiment. And the discussions are as follows.

In terms of device noise of memristor, the inherent errors in memristor weight mapping can accumulate across layers in multi-layer neural networks, with more pronounced effects in deeper networks. The data-in-situ computing network applies weight vectors to image for computing, which actually flattens the network depth. So, it can reduce the error propagation across layers.

Regarding device endurance, we propose a One Pulse Method for device modulation, which can significantly reduce the number of operational pulses and avoid repeated write-erase cycles. This method will contribute to extending the lifespan of memristor devices, providing crucial assurance for practical applications.

In terms of scalability, the data-in-situ computing network eliminates the need for full-image input in conventional architectures. This not only substantially reduces the scale requirements for subsequent classification networks but also decreases the dependency on memristor array size. So, this architecture provides great foundation for building larger-scale visual processing systems in the future.

To clarify this, we have added an overall hardware picture in Figure 1, and provided a detailed introduction of the 1PnR hardware implementation and image acquisition experiment, and **modified the manuscript (Section 2.2, Paragraph 3,4, Highlighted in yellow)** as: “Utilizing the fabricated pixel sensors and transmission circuit, a 1PnR hardware verification system is implemented, as shown in **Supplementary Fig. S6**. UV LED light sources are employed, corresponding one-to-one with the fabricated pixel sensors. The image is converted into UV light signals based on pixel values and directly irradiated to the optical sensors, which simulates the optical exposure process. And the detailed process of the exposure experiment based on the hardware system is illustrated in **Supplementary Fig. S7**. Besides, to achieve RES verification with larger image size based on the prepared image acquisition hardware, the target column of image (up to 30 pixels) is reshaped to a 5×6 array for UV irradiation. The light signals received by the optical sensors are then reshaped back to one column through the 30-channel conversion circuits and then written to target column of the connected memristor array, which is shown in **Supplementary Fig. S8**.”

Correspondingly, we have modified the Figure 1, Supplementary Fig. S6, S8 as:

Fig. 1. Schematic of the human visual system and the proposed artificial visual system based on 1PnR structure. The human vision system contains eyes, optic neurons and brain, the visual working memory in human brain stores temporary sequential visual information and performs pre-process. The proposed artificial visual system based on 1PnR architecture is composed of ITO/ZnS/TiN pixel sensors for image sensing, analogue circuit for data transmission, and an 8k memristor array based on TiN/HfO₂/TaO_x/TiN stack. The array is divided into 2 parts, one part serves as working memory, which stores image data and performs data-in-situ for pre-process. The other part serves as neuromorphic computing core, which stores network weights and carries network computing for classification.

Supplementary Figure 6. Hardware implementation of the 1PnR architecture. The UV LED array board and UV sensor array board are positioned and fixed by one-to-one for light emitting and sensing per pixel. Opaque grids are applied to avoid cross-talk between light signals of different pixels. The 5×6 optical sensors are then connected to 30 channels of transmission circuit, which converts light information to gate voltages of memristor array for further image storage. The memristor modulation board mainly consists of FPGA core controller, array read-write circuits (ADC&DAC), WL connecting ports and switch matrix. Each WL port offers analogue connection to 30 channels of the 8k memristor array's WLs through switch matrix. And switch matrix can switch the connection of the memristor array' WLs between the WL ports for image exposure or DAC chips for data-in-situ computing.

Supplementary Figure 8. The Schematic diagram of light collection mechanism for rolling exposure experiment. Each column of the target image is reshaped to 5×6 array for exposure experiment. The received signals are reshaped back to one column through the 30-channel conversion circuits and further written to the corresponding column of memristor array in parallel using the proposed One-Pulse Method. Then, next column of image is exposed and stored to the corresponding next column, till the whole image is sensed and stored.

Corresponding change in manuscript: Yes

Location of Change:

Section 2.2: One-Pixel-Multiple-Memristor Structure and Rolling exposure strategy for fast image acquisition

Page 11:

Paragraph 3 of Section 2.2

Figure 1

Supplementary Figure 6

Supplementary Figure 8

Comment #4:

Reviewer wrote:

Although the architecture is described as bio-inspired, the biological underpinnings—particularly the function and mechanism of visual working memory in the human visual system—are only briefly introduced. Providing a more thorough explanation of how specific components of the proposed system (e.g., sensor, memory, and compute modules) correspond to biological elements such as the retina, visual cortex, and working memory would enhance the conceptual grounding and clarify the biomimetic relevance.

Our response:

Thanks for your comment. We agree that a deeper elaboration of the biological parallels is essential to fully ground our work in neuromorphic principles. Our proposed 1PnR architecture is not merely inspired by biology; it is a deliberate physical emulation of the functional hierarchy and computational principles of the primate visual system, as shown in Figure R18.

Figure R18. The Imation of human visual system and the 1PnR architecture and system

Below, we provide a detailed point-by-point correspondence between the biological elements and our hardware implementation, also depicted in Figure R19.

1. Retina → ITO/ZnS/TiN Pixel Sensors

Biological Mechanism: The retina performs phototransduction, where photoreceptors (rods and cones) convert

light into electrochemical signals, incorporating initial preprocessing like adaptation.

Hardware Correspondence & Embodiment: The pixel sensor is achieved by ITO/ZnS/TiN photo-resistive structure. ZnS functional film with excellent photoactivity is engineered to mimic this photoconversion. The experimental result shows the device has great C2C and D2D uniformity, with the ability to sense the light pulse with various intensities confirms its role as a reliable, biomimetic sensory layer ^[1].

2. Optic Nerve & Neural Pathways → Analog Transmission Circuits

Biological Mechanism: The optic nerve axons transmit encoded visual information via action potentials. This transmission is analog in nature and highly energy-efficient.

Hardware Correspondence & Embodiment: The analog circuits for conversion and transmission of sensory data fulfill this role. They connect the sensors to the memristor array, transmitting converted voltage signals without analog-to-digital conversion, thus mimicking the efficient, lossy transmission of biological neural pathways.

3. Visual Working Memory (Prefrontal Cortex) → Non-Volatile Memristor Working Memory Array

Biological Mechanism: The visual working memory (VWM) in the prefrontal cortex (PFC) maintains a short-term, low-power representation of visual information through sustained neuronal firing activity. This active retention allows for manipulation and preprocessing of data (e.g., tracking, comparison) without constant recall from long-term storage ^[2, 3].

Hardware Correspondence & Embodiment: This is the core biomimetic innovation. A portion of our 8K memristor array is designated to serve as working memory, which stores image data and performs data-in-situ for pre-process. The critical link is that the non-volatility of the memristor—its ability to maintain its conductance state without power—is the direct physical analogue to sustained neuronal firing. This enables the stored image data to be kept active with minimal energy, precisely mimicking the "active retention" function of biological VWM. The data-in-situ preprocessing, where information is processed where it is stored, eliminates energy-intensive data movement and is a direct hardware implementation of the integrated memory-processing observed in the PFC.

4. Visual Cortex (V1/V2) → Memristor-based Neuromorphic Computing

Biological Mechanism: The primary visual cortex (V1) extracts basic features (edges, orientations) via complex convolutional operations, while higher areas perform object recognition and classification ^[4].

Hardware Correspondence & Embodiment: The other part of our memristor array serves as neuromorphic computing core, which stores network weights and carries network computing for classification. The 1T1R memristor array performs matrix multiplication in the analog domain via Ohm's law and Kirchhoff's current law. This mimics the hierarchical feature extraction of the visual cortex. The analogue switching behavior enable this dense, efficient parallel computation, emulating the adaptive synaptic weights in biological neural networks.

5. Retinal Divergent Connectivity → 1PnR (One-Pixel-Multiple-Memristor) Structure

Biological Mechanism: A single photoreceptor connects to multiple bipolar cells, fanning out information in parallel for simultaneous processing of different visual features ^[5].

Hardware Correspondence & Embodiment: The 1PnR Structure is a direct anatomical mimic of this. Here, each pixel sensor is connected to a gate line of the 1T1R array, allowing a single channel of optic input to (be stored to) multiple memristor cells simultaneously. This enables the rolling exposure strategy where an image is stored column by column with high parallelism.

References:

- [1] Masland, R. H. (2012). The neuronal organization of the retina. *Neuron*, 76(2), 266–280.
- [2] Fuster, J. M. (2015). *The Prefrontal Cortex* (5th ed.). Academic Press.
- [3] Baddeley A D. Working memory[J]. *Philosophical Transactions of the Royal Society of London. B, Biological Sciences*, 1983, 302(1110): 311-324.
- [4] Hubel, D. H., & Wiesel, T. N. (1962). Receptive fields, binocular interaction and functional architecture in

the cat's visual cortex. *The Journal of Physiology*, 160(1), 106–154.

[5] Pasternak, T., & Greenlee, M. W. (2005). Working memory in primate sensory systems. *Nature Reviews Neuroscience*, 6(2), 97–107.

Figure R19. The correspondence between the biological elements and 1PnR hardware implementation.

To clarify this, we have modified the manuscript (Section 1, Paragraph 3, Highlighted in yellow) as: In this work, inspired by the high-efficiency working mode of the visual nerves in the human system, we propose new neuromorphic visual architecture, namely, a one-pixel-74 multiple-memristor (1PnR) computing architecture for sequential image sensing, storing and processing, as shown in Fig. 1. The light sensors and memristor array, similar to biological retina, are connected by analog circuits as optic nerves for conversion and transmission of the sensory data. The array is divided into 2 parts, one part serves as visual working memory, which stores image data and performs data-in-situ for pre-process. The other part serves as neuromorphic computing core mimicking visual cortex, which stores network weights and carries network computing for classification. Furthermore, the sensors and memristor array are integrated by a gate multiplexing architecture, where a sensor is connected to a gate line of the 1T1R array, offering the ability of storing a single channel of optic input to multiple memristor cells simultaneously, functioning as retinal divergent connectivity for visual information processing. Based on the structure, a rolling exposure strategy for fast sequential image acquisition is then proposed.

Corresponding change in manuscript: Yes

Location of Change:

Section 1: Introduction

Page 4:

Paragraph 3 of Section 1

Comment #5:

Reviewer wrote:

Despite the authors performed the classification of binary(black and white) images, there is limited discussion regarding the handling of color images. Given that the memristor-based system supports multiple resistance state, it would be valuable to evaluate whether this architecture can maintain performance when dealing with color images. Further investigation into this aspect would strengthen the paper's applicability to a broader range of real-world image processing task.

Our response:

Thank you for your comments. Due to the format limitations of the Weizmann dataset, binary images are used in the human activity recognition experiment. The Weizmann dataset is employed to evaluate the recognition of sequential-frame images. Additionally, we experimentally tested the MNIST dataset using grayscale images, leveraging the multi-level characteristics of the memristor. The experimental results are presented in Supplementary Figure 13.

As for the color image, acquiring color images requires multi-color sensors, which have not yet been implemented in our current work. To validate the capability of the data-in-situ computing network in recognizing color image, we selected the SVHN color dataset for simulation. It should be noted that since the SVHN dataset contains more complex images, extracting local features is essential for achieving better recognition performance. The data-in-situ computing can also extract local features by shorting the voltage weight vectors and applying sliding operations, like conventional computing. Thus, we propose a data-in-situ convolutional computing network (DICNN) architecture, as illustrated in Figure R20. Small-sized weight voltage vectors are applied to the memristor array storing the image for data-in-situ computation, followed by a sliding-window operation, similar to convolutional computation. And a feature map can be finally obtained.

Figure R20. The schematic diagram of data-in-situ conventional computing. Small-sized weight voltage vectors are applied to the memristor array storing the image for data-in-situ computation, followed by a sliding-window operation, similar to convolutional computation. A feature map can be finally obtained.

Based on data-in-situ convolutional computing principle, we preliminarily validated the data-in-situ convolutional network architecture for the SVHN dataset, as shown in Figure R21a. The RGB channels of the color image are simulated to store in three separate memristor arrays, and each array is processed using a 1×3 data-in-situ convolutional kernel to generate three-channel feature maps. These features are then fed into a network comprising

two convolutional layers and pooling layers for further classification. The simulation results of the network are shown in Figure R21b, where the in-situ convolutional network achieves a recognition rate of 95.97%, which is comparable to that of a traditional CNN with the same structure (conv1: $3 \times 3 \times 3$, conv2: $3 \times 3 \times 3 \times 8$, MaxPool1: 2×2 , conv3: $3 \times 3 \times 3 \times 16$, MaxPool2: 2×2 , FC: $1024 \times 128 \times 10$) shown in Figure R21c.

The data-in-situ convolutional network is part of our ongoing work. Here, we present our preliminary simulation results, which demonstrate that the data-in-situ computing paradigm exhibits the capability to process color images and more complex visual data, indicating its strong generalization potential.

Figure R21. The data-in-situ conventional network for SVHN dataset a the schematic diagram of the data-in-situ convolutional network architecture for the SVHN dataset. A 1×3 data-in-situ convolutional kernel is applied to 3 memristors storing RGB channels of the color image to generate three-channel feature maps. The feature maps are then fed into a network comprising two convolutional layers and pooling layers for further classification. b simulation results of the data-in-situ convolutional network for SVHN dataset, which achieves a recognition rate of 95.97%. c simulation results of same structure traditional convolutional network for SVHN dataset as reference, a 96.15% accuracy demonstrate that the data-in-situ computing paradigm exhibits the capability to process color images and more complex visual data

Corresponding change in manuscript: No

Reviewer #3 (Remarks to the Author):

This manuscript, “Data-In-situ Computing with One-Pixel-Multiple-Memristor Architecture for 1 Neuromorphic Sequential Vision,” is inspired by the working memory mechanism of the human visual system and proposes a novel neuromorphic visual architecture with a single-pixel multiple-memristor (1PnR) structure. Additionally, the authors propose a rolling exposure strategy based on open-loop single-pulse modulation, which leverages the array architecture of the 1PnR to enable column-by-column sensing of images while simultaneously storing signals. This strategy enhances the system's temporal and energy efficiency. Finally, the authors propose a resistive-based in-situ data computation network distinct from traditional resistive networks, mapping voltage

signals as weights and conductance as image data. This method eliminates the need for physical transmission of image data, instead performing computations directly, significantly improving system efficiency. Finally, the authors constructed a 1PnR hardware system comprising 30 optical sensors and an 8k29 memristor array, achieving an image recognition rate of 95.27% on the Weizmann Human Action Flow Dataset. This work is highly innovative, with the proposed strategies and algorithms effectively addressing issues faced by traditional methods, and holds significant value in applications such as image recognition, processing, and analysis. However, there are still some problems that need to be solved in this work.

Comment 1:

Reviewer wrote:

There are two errors in the image descriptions. Please confirm and correct them if they are indeed errors: First, the reference to Figure 2f in line 160 of the manuscript should refer to Figure 2g; second, the reference to Figure 2g in line 234 of the manuscript should refer to Figure 3g.

Our response:

We sincerely thank the reviewer for their meticulous reading and for identifying these citation errors. The reference in line 160 has been corrected from "Fig. 2f" to "Fig. 2g". The reference in line 234 has been corrected from "Fig. 2g" to "Fig. 3g". **We have carefully checked the entire manuscript for similar inconsistencies and corrected them accordingly.**

Corresponding change in manuscript: Yes

Location of Change:

Section 2.1: Device Characterization and One-Pulse Modulation Method

Page 8:

Paragraph 3 of Section 2.1

*Section 2.2: One-Pixel-Multiple-Memristor Structure and Rolling exposure strategy
for fast image acquisition*

Page 12:

Paragraph 5 of Section 2.2

Comment 2:

Reviewer wrote:

There is a problem with the subscript in image 2d. Please correct it.

Our response:

Thank you for your careful review and pointing out this mistake for us. The subscript in Figure 2d has been corrected to the correct number.

Corresponding change in manuscript: Yes

Location of Change:

Figure 2d

Comment 3:

Reviewer wrote:

The fonts in the image are inconsistent. Please correct them. In addition, the image is not clear enough. Please replace it with a high-definition image.

Our response:

Thank you for your comments. **We have modulated the figures using a unified and professional font (Arial) throughout to ensure consistency. Furthermore, all figures have been replaced with high-resolution versions (300 dpi) to ensure clarity and sharpness in the revised manuscript.**

Corresponding change in manuscript: Yes

Location of Change:

All the Figures

Comment 4:

Reviewer wrote:

The manuscript only shows optical images of the array and TEM images of the memristors. Please add a diagram of the device structure of a single unit in the array (a schematic diagram of the structure connecting the transistor and the memristor).

Our response:

Thank you for your suggestion. In a 1T1R unit, the memristor's bottom electrode is connected to the drain of the transistor, the schematic diagram of the 1T1R structure is shown in Figure R22. The top electrode of the memristor is connected to the RL of the 1T1R array, while the gate and source of the transistor are connected to WL and SL of the array, respectively. The transistor can provide current compliance to the attached memristor in resistance modulation and serve as a current switch to suppress current sneak-path in the 1T1R array.

Figure R22. The schematic diagram of the 1T1R structure. The memristor's bottom electrode is connected to the drain of the transistor. The top electrode of the memristor is connected to the RL of the 1T1R array, while the gate and source of the transistor are connected to WL and SL of the array, respectively.

To provide greater clarity on the 1T1R architecture, **we have modified the manuscript (Section 2.1, Paragraph 1, Highlighted in yellow) as: “And the schematic diagram of the 1T1R structure as well as the transistor's output characteristic are presented in Supplementary Fig. S1”, and added the schematic diagram of the 1T1R structure to Supplementary Figure 1 as:**

Supplementary Figure 1. **a** schematic diagram of the structure connecting the transistor and the memristor. The memristor's bottom electrode is connected to the drain of the transistor. The top electrode of the memristor is connected to the RL of the 1T1R array, while the gate and source of the transistor are connected to WL and SL of the array, respectively. The transistor can provide current compliance to the attached memristor in resistance modulation and serve as a current switch to suppress current sneak-path in the 1T1R array. **b** the output characteristic of the transistor in the 1T1R array. The transistor has different current compliance with various gate voltages (from 1 V to 4 V with 0.5 V step).

Corresponding change in manuscript: Yes

Location of Change:

Section 2.1: Device Characterization and One-Pulse Modulation Method

Page 5:

Paragraph 1 of Section 2.1

Supplementary Figure 1

Comment 5:

Reviewer wrote:

The article presents the electrical characteristics of memristors and individual units, but does not present the electrical characteristics of transistors. Please supplement the electrical characteristics of transistors in the array, such as transfer characteristic curves and output curves, to further prove the principle of array function implementation.

Our response:

Thanks for your comment. As suggested, we have conducted electrical test of the transistor to prove the principle of array function implementation. The output characteristic of the transistor is plotted in Figure R23. The output characteristic curves (I_{ds} - V_{ds}) show the transistor's behavior under different gate voltages (from 1 V to 4 V with 0.5 V step), which supports the current compliance of the attached memristor in the resistance modulation.

Figure R23. The output characteristic of the transistor in the 1T1R array. The transistor has different current compliance with various gate voltages (from 1 V to 4 V with 0.5 V step in the experiment), which supports the attached memristor in resistance modulation.

To clarify this, we have modified the manuscript (Section 2.1, Paragraph 1, Highlighted in yellow) as: “And the schematic diagram of the 1T1R structure as well as the transistor’s output characteristic are presented in Supplementary Fig. S1”, and added the schematic diagram of the 1T1R structure to Supplementary Figure 1 as:

Supplementary Figure 1. a) schematic diagram of the structure connecting the transistor and the memristor. The memristor’s bottom electrode is connected to the drain of the transistor. The top electrode of the memristor is connected to the RL of the 1T1R array, while the gate and source of the transistor are connected to WL and SL of the array, respectively. The transistor can provide current compliance to the attached memristor in resistance modulation and serve as a current switch to suppress current sneak-path in the 1T1R array. b) the output characteristic of the transistor in the 1T1R array. The transistor has different current compliance with various gate voltages (from 1 V to 4 V with 0.5 V step).

Corresponding change in manuscript: Yes

Location of Change:

Section 2.1: Device Characterization and One-Pulse Modulation Method

Page 5:

Paragraph 1 of Section 2.1

Supplementary Figure 1

Comment 6:

Reviewer wrote:

Supplemental Figure 1, the unit of light intensity in the 17th row of the image description does not match that in Supplemental Figure 1b.

Our response:

Thanks for your comment. The unit of light intensity in the 17th row is a mistake, which should be $\text{pW} \cdot \mu\text{m}^{-2}$. Thanks for pointing out this mistake for us, **we have corrected in the revised manuscript.**

Corresponding change in manuscript: Yes

Location of Change:

Supplemental Figure 2 of the revised manuscript

Comment 7:

Reviewer wrote:

The supplementary figures 1b and 1c show changes in voltage as well as switching of light conditions, which do not clearly demonstrate the performance of the light sensing unit. Please supplement the relevant measurements to prove the performance of the light sensor more rigorously.

Our response:

Thank you for your comment. To more comprehensively validate the performance of the sensing unit, we have conducted dynamic response tests of the conversion circuit, as shown in Figure R24a.

Figure R24. the transition characteristic of the transmission circuit in the sensing unit. a the experiment setup of transmission speed test. A signal generator is connected to the input ports of the differential amplifier to provide the reference voltage V_{ref} and the voltage division V_{OR} on the optical sensor, while the oscilloscope simultaneously monitors the V_{OR} and the output voltage V_{out} of the circuit. **b** The experiment result of transmission speed test, indicating ~920 ns transmission time.

A simplified model of the conversion circuit is presented, and the speed is tested by a signal generator and an oscilloscope. The signal generator is connected to the input ports of the differential amplifier to provide the reference voltage V_{ref} and the voltage division V_{OR} on the optical sensor, while the oscilloscope simultaneously monitors the

V_{OR} and the output voltage V_{out} of the circuit. The captured waveform of the speed test is shown in Figure R24b. The yellow curve represents the voltage division V_{OR} of the optical sensor, the falling portion is focused because V_{OR} drops under UV illumination. The green curve represents the output voltage of the differential circuit, which rises due to the decrease in the negative input. The results indicate that the time from the input voltage drop to the stabilization of the output voltage rise is about 920 ns, which reflects the speed of our conversion circuit. It should be noted that this delay is measured at the board level. With future chip-level integration, where parasitic capacitance and other non-ideal effects can be greatly reduced, we believe the circuit speed can be further improved.

To clarify this, we have **modified the manuscript (Section 2.2, Paragraph 1, Highlighted in yellow)** as: “The circuit parameter is adjusted to fit the gate voltage range of the 1T1R array, and the transmission relationship of the circuit from light density to gate voltage, as well as the transmission speed is plotted in **Supplementary Fig. S5.**” and the Supplementary Fig. S5 as:

Supplementary Figure 5. a the relationship of the transmission circuit from light density to output voltage. The black spot is the measured data, and the purple curve is the fitted result with power function relationship: $I = a \cdot Db + c$. The circuit process is purely linear, so the fitted relationship is consistent with the sensor’s characteristic shown in Supplementary Fig. S2d. Furthermore, the output voltage of the circuit is adjusted to fit gate voltage of the 1T1R cell. **b The responding speed of the transmission circuit.** A signal generator is connected to the input ports of the differential amplifier to provide the signal, while an oscilloscope simultaneously monitors the input and output signal of the circuit. The yellow curve represents the voltage division V_{OR} of the optical sensor, the falling portion is focused because V_{OR} drops under UV illumination. The green curve represents the output voltage of the differential circuit, which rises due to the decrease in the negative input. The results indicate that the time from the input voltage drop to the stabilization of the output voltage rise is about 920 ns, which reflects the speed of our conversion circuit. It should be noted that this delay is measured at the board level. With future chip-level integration, where parasitic capacitance and other non-ideal effects can be greatly reduced, we believe the circuit speed can be further improved.

Corresponding change in manuscript: Yes

Location of Change:

Section 2.2: One-Pixel-Multiple-Memristor Structure and Rolling exposure strategy for fast image acquisition

Page 9:

Paragraph 1 of Section 2.2

Supplementary Figure 5

Comment 8:

Reviewer wrote:

Please explain the RS window in detail.

Our response:

Thank you for your comments.

The Resistive Switching (RS) window is defined explicitly as the ratio between the High Resistance State (HRS) and the Low Resistance State (LRS), often expressed as HRS/LRS. A larger RS window is critical as it can enhance the readout margin between different states, improving the reliability of data storage and the accuracy of data-in-situ computation. Also, it makes the device more resilient to noise and variations in the operating conditions.

The RS window can be extracted from the IV curve of the memristor. Figure R25 shows the quasi-DC characteristics of TiN/TaO_x/HfO_x/TiN memristor. During positive voltage sweep, the device switches from HRS to LRS, while during negative sweep, it transitions from LRS back to HRS. Using the responding current at 0.1 V as a reference, the device exhibits a current level of $\sim 10^{-6}$ A in the HRS and $\sim 10^{-5}$ A in the LRS, resulting in a resistive switching window of approximately 10.

Figure R25. The IV curves of the 1T1R memristor. Under 0.1 V monitoring voltage, the device exhibits a current level of $\sim 10^{-6}$ A in the HRS and $\sim 10^{-5}$ A in the LRS, resulting in a resistive switching window of approximately 10.

To clarify this, we have modified the manuscript (Section 2.1, Paragraph 1, Highlighted in yellow) as: “With a 1.6 V gate voltage in SET voltage sweep (0→1.0 V) and 3.5 V gate voltage in RESET sweep (0→-1.15 V), a Resistive Switching (RS) window, defines as the ratio between the High Resistance State (HRS) and the Low Resistance State (LRS), can reach to ~ 10 ”.

Corresponding change in manuscript: Yes

Location of Change:

Section 2.1: Device Characterization and One-Pulse Modulation Method

Page 5:

Paragraph 1 of Section 2.1